



# Shifts in organic matter character and microbial community structure from glacial headwaters to downstream reaches in Canadian Rocky Mountain rivers

Hayley F. Drapeau[1,2], Suzanne E. Tank[1], Maria Cavaco[2], Jessica A. Serbu[1], Vincent L. St.Louis[1], Maya P. Bhatia[2,3]

[1]Department of Biological Sciences, University of Alberta, Edmonton, T6G 2R3, Canada
[2]Department of Earth and Atmospheric Sciences, University of Alberta, Edmonton, T6G 2R3, Canada
[3]Deceased
*Correspondence to*: Suzanne Tank (suzanne.tank@ualberta.ca)

**Abstract.** Climate change is causing mountain glacial systems to warm rapidly, leading to increased water fluxes and concomitant export of glacially-derived sediment and organic matter (OM). Glacial OM represents an aged, but potentially bioavailable carbon pool that is compositionally distinct from OM found in non-glacially sourced waters. Despite this, the composition of riverine OM from glacial headwaters to downstream reaches and its role in structuring microbial communities has yet to be characterized in the Canadian Rockies. Over three summers (2019–2021) we collected samples before, during and after glacial ice melt along stream transects ranging 0 – 100 km downstream of glacial termini on the eastern slopes of the Canadian Rocky Mountains. We quantified dissolved and particulate organic carbon (DOC, POC) concentration, and used isotopes ($\Delta^{14}$C-OC, $\delta^{13}$C-OC) and dissolved OM (DOM) absorbance and fluorescence to assess OM age, source, and character. Environmental data were combined with microbial 16S rRNA gene sequencing to assess controls on microbial community composition. From glacial headwaters to downstream reaches, OM showed a clear transition from being aged and protein-like with an apparent microbial source to being relatively younger and humic-like. Indicator microbial species for headwater sites included chemolithoautotrophs and taxa known to harbour adaptations to cold temperatures and nutrient-poor conditions, suggesting a role for glacial seeding of microbial taxa to headwaters of this connected riverine gradient. However, environmental conditions (such as deuterium excess, an indicator of water source; water temperature; POC concentration; and protein-like DOM) could only significantly explain ~9% of the observed variation in microbial community structure. This finding, paired with the identification of a ubiquitous core microbial community that comprised a small proportion of all identified amplicon sequence variants (ASVs), but was present in large relative abundance at all sites, suggests that mass effects largely overcome species sorting to enable a connected microbial community along this strong environmental gradient. Thus, with a loss of novel glacial and microbial inputs with climate change, our findings suggest consequent changes in OC cycling and microbial community structure may lead to complex ecosystem responses across the evolving mountain-to-downstream continuum in small, glacierized systems.



# 1 Introduction

Anthropogenic climate change is causing glaciers to retreat at unprecedented rates (Zemp et al., 2015). Rapid glacial retreat has the potential to impact downstream hydrology (Clarke et al., 2015), carbon cycling (Hood et al., 2015; Hood et al., 2020) and microbial community dynamics (Bourquin et al., 2022; Hotaling et al., 2017), as source water contributions to headwater
streams undergo fundamental change. In the Canadian Rocky Mountains, glacial meltwater contributions to downstream rivers is presently increasing but is projected to peak by 2040 (Clarke et al., 2015; Pradhananga and Pomeroy, 2022). Over the coming decades, the eventual disappearance of glacial meltwater additions to glacially-fed rivers is predicted to lead to decreased summer flows and water availability, impacting up to one million individuals in communities downstream (Anderson and Radić, 2020).

Changing contributions of glacial meltwater to headwater streams could disrupt downstream riverine carbon dynamics, because glaciers provide, and internally cycle, unique sources of organic carbon (OC) (Hood et al., 2015; Musilova et al., 2017; Wadham et al., 2019). As glaciers melt, the formation of englacial and marginal channels can cause meltwater to be routed on top of, through, and/or underneath, glaciers (Nienow et al., 1998). At the beginning of the melt season, water is
typically sourced from snowmelt which is inefficiently routed through a distributed hydrological network at the ice–bed interface or along the ice margin, and is characterized by high rock:water contact times (Arendt et al., 2016). As the melt season progresses an efficient subglacial network forms facilitating rapid transit through the subglacial environment (Arendt et al., 2016) or routing of water laterally via marginal streams. This meltwater is then exported into headwater streams where it provides an important seasonal control on stream hydrology via augmentation of summer discharge (Campbell et al., 1995).
The contribution of glacially sourced water to headwater streams enables movement of novel organic matter (OM), from overridden vegetation at the glacier bed (Bhatia et al., 2010), anthropogenic aerosols on the ice surface (Stubbins et al., 2012), and in situ microbial communities (Stibal et al., 2012) to downstream systems. Previous studies have shown that glacially derived dissolved OM (DOM) is aged (Bhatia et al., 2013; Hood et al., 2009; Stubbins et al., 2012), exhibits protein-like fluorescence (Barker et al., 2009; Kellerman et al., 2020), and can serve as a labile substrate for downstream microorganisms
(Hood et al., 2009; Singer et al., 2012). In contrast, glacially derived particulate OM (POM, and its OC subset POC), which has been less studied, is thought to be derived largely from comminuted rock and sediment, and to persist within rivers (Hood et al., 2020). During transit through fluvial networks, glacially exported OM mixes with non-glacial terrestrially derived OM, which typically exhibits a humic-like fluorescent signature (McKnight et al., 2001a), is relatively younger (Raymond and Bauer, 2001), and has generally been shown to be less accessible for microbial consumption (D'Andrilli et al., 2015).

River ecosystems are overwhelmingly heterotrophic, with food webs sustained by microbial consumption and mineralization of OM inputs (Bernhardt et al., 2022). OM composition has been found to be an important determinant of microbial community structure (Judd et al., 2006) alongside environmental factors such as light, river flow regimes (Bernhardt et al., 2022; Milner





et al., 2017), temperature, and nutrient availability (Elser et al., 2020). Thus, the increase and eventual decline of glacial

meltwater inputs to fluvial networks, with their entrained glacial OM, may impact microbial community structure. In addition, headwater community structure also shift as a result of the direct loss of novel glacially-sourced microbes (Wilhelm et al., 2013) as glaciers have been found to host unique microbial taxa in both the supraglacial and subglacial environments (Bourquin et al., 2022; Hotaling et al., 2017; Stibal et al., 2012). These taxa are often uniquely adapted to glacial conditions, such as seasonally fluctuating and overall low nutrient concentrations, cold temperatures, and the presence of reduced chemical species

in low oxygen subglacial environments, which can select for chemosynthetic metabolisms (Bourquin et al., 2022; Hotaling et al., 2017). Recent work at European, Greenlandic and North American glaciers has found that surface meltwaters contained a substantial number of microbial cells (on average of $10^4$ cells mL$^{-1}$), highlighting the potential of supraglacial environments to seed downstream microbial communities (Stevens et al., 2022). Overall, if changes in glacial loss drive changes in microbial community composition, this could lead to a loss of unique metabolic pathways (Bourquin et al., 2022), changes in respiratory

efflux of $CO_2$ (e.g., Singer et al., 2012), and food web perturbations, given that OM incorporated into microbial biomass is assimilated by higher trophic level organisms (e.g., Fellman et al., 2015).

In this study, we pair measurements of stream OM and microbial community composition along transects from glacial headwaters to downstream rivers, with sampling across three years and encompassing conditions prior to, during, and after the

summer glacial melt season. We undertake this work in glacially-fed fluvial networks draining eastward from the Canadian Rocky Mountains, to assess how glacial loss will impact downstream microbial diversity and carbon cycling in this region. The icefields (Wapta and Columbia) that feed these alpine glaciers are rapidly shrinking and are projected to be reduced by 80–100% of their 2005 area by 2100 (Clarke et al., 2015). Due to the increasing rates of glacier loss, glacier meltwater inputs to rivers are currently increasing (Pradhananga and Pomeroy, 2022), but are projected to peak at some point before 2040

(Clarke et al., 2015). Our objectives in this rapidly changing landscape were three-fold: first, to investigate spatial and temporal variation in stream OM age, source, and character across different hydrological periods in three Rocky Mountain rivers with a distance ranging from 0–100 km downstream from glacial termini; second, to characterize microbial community composition and its variation along this same gradient; and finally, to identify environmental drivers of variation in microbial community structure, with a particular focus on OM composition.  Despite strong gradients in OM character, apparent source, and age

along our 100 km transects, we find overall cohesion in microbial community structure, indicating that mass effects may be an over-riding control on community dynamics within these connected fluvial networks. At the same time, however, a clear presence of glacially associated microbial taxa suggests that glaciers are seeding microbial communities in the glacial headwaters, reinforcing the critical nature of the cryosphere microbiome in poorly studied mountain glacial systems.



## 2 Methods

### 2.1 Site description

Samples were collected from the Athabasca–Sunwapta, North Saskatchewan (NSR) and Bow rivers located within Banff and Jasper National Parks of the Canadian Rocky Mountains (Fig. 1). Each of these rivers are sourced from glacial headwaters with the Athabasca–Sunwapta and NSR receiving inputs predominately from the Columbia Icefield, and the Bow River receiving inputs from the Wapta Icefield. At ~216 km$^2$ (Tennant and Menounos, 2013; Bolch et al., 2010), the Columbia Icefield is the largest icefield in the Canadian Rockies, while the Wapta Icefield is smaller at ~80 km$^2$ (Ommanney, 2002).

Along each river we chose three to four sampling sites at which we examined how stream OM and microbial characteristics evolved with increasing distance from glacial input. Sampling sites ranged from glacial headwaters (0 km downstream of glacial inflow) to 100 km downstream of glacial termini. For certain statistical analyses, sites were binned into four distance ranges: headwater (0.3 km downstream), near (3–7 km downstream), mid (18–35 km downstream) and far (40–100 km downstream) (Fig. 1a; Table S1). Sites were restricted to locations within Banff and Jasper National Parks to enable a comparison of study sites that were minimally impacted by direct anthropogenic landscape disturbance. Throughout, we generally use "stream" to refer to specific sites, although we acknowledge that our transects span from glacial headwaters to relatively large riverine reaches.

Samples were collected over a three-year study period (Fig. 1b). Each of the sample sites were visited every three to four weeks throughout the months of May to October during 2019, 2020, and 2021. Opportunistic samples were also collected at a subset of sites in December 2019 and January 2021. This sampling design enabled us to cover the three main hydrological stages in a glacially sourced river: prior to the seasonal glacial ice melt (pre-melt), the glacial melt period (melt), and the post-glacial ice melt period (post-melt). Hydrological periods were determined using the Environment and Climate Change Canada (ECCC) stream discharge hydrographs from Athabasca Glacier (station ID: 07AA007) (ECCC, 2022) (Fig. 1b). We defined the pre-melt stage as the period when monthly average stream discharge at the Athabasca glacier headwater was low (< 1 m$^3$ s$^{-1}$); a likely indication that glacial meltwater channels were not yet established (Arendt., 2015). The glacial melt period was defined as the period when monthly average discharge was high (>1, but rapidly transitioning to >2.5 m$^3$ s$^{-1}$; discharge typically peaked during July–August); at this time glacial meltwater channels were likely well established, with melting glacier ice contributing substantially to headwater stream flow. Finally, the post-melt period was characterized by, once again, low average monthly discharge at the glacier headwater (< 1 m$^3$ s$^{-1}$; typically, October and onwards); likely indicating closed glacial channels and cessation of ice melt input. In 2019, all three hydrological stages (pre-melt, melt, post-melt) were sampled; in 2020, sampling was delayed due to the COVID-19 pandemic, and samples were only collected during the melt and post-melt





stages; in 2021, samples were collected during the pre-melt and melt stages only. Over our three-year study period, 2019
exhibited the lowest melt-period discharge and 2021 the highest (Fig. 1b).

## 2.2 Field sampling and field laboratory processing

At each site samples were collected for: DOM absorbance and fluorescence, DOC concentration, POC concentration,
particulate and dissolved OC isotopes ($\delta^{13}$C-DOC, $\delta^{13}$C-POC, $\Delta^{14}$C-DOC, $\Delta^{14}$C-POC), and 16S rRNA gene sequencing. To

further assess water source and hydrochemical controls on microbial diversity in our analyses, we also sampled for water
isotopes ($\delta^{18}$O-H$_2$O, $\delta^2$H-H$_2$O), nutrients (ammonia (NH$_4^+$), nitrate and nitrite (NO$_3^-$+NO$_2^-$), total dissolved nitrogen (TDN),
total nitrogen (TN), soluble reactive phosphorous (SRP), total phosphorous (TP), and dissolved silica (dSi)), major anions
and cations (Mg$^{2+}$, Cl$^-$, Na$^{2+}$, SO$_4^{2-}$, Ca$^{2+}$, K$^+$), trace metals (Al, Ba, Cr, Mn, Mo, Ni, Sr), temperature, and specific
conductance. All sample bottles, except those for anions and nutrients, were soaked overnight in a dilute acid bath (1.2 mol

L$^{-1}$ trace metal grade HCl), rinsed at least 3 times with 18.2 MΩ MilliQ water, and rinsed three times with sample water
before collection at the field site. Trace metal bottles were pre-soaked in a 0.01% Citranox bath prior to the acid bath step.
Following the HCl soak and MilliQ water rinse, all glassware (bottles, EPA vials, and filtration apparatus) were combusted
at 560°C for a minimum of 4 h, and high-density polycarbonate (HDPC) bottles for microbial analysis were autoclaved.
Glass microfibre filters (grade GF/F, Whatman) were combusted at 460°C for a minimum of 4 h prior to use. High-density

polyethylene (HDPE) bottles for ion collection were pre-washed with Citranex prior to the acid cleaning procedure.

Samples for DOM absorbance and fluorescence, DOC concentration and $\delta^{13}$C-DOC were sub-sampled from a 250 mL amber
glass collection bottle and filtered streamside through 0.45 μm polyethersulfone (PES) filters (Fisherbrand Basix; pre-rinsed
with 60 mL Milli-Q and 15 mL river water) into combusted 40 mL amber EPA vials. Nutrient samples were sub-sampled from

a 500 mL HDPE collection bottle and filtered streamside using a pre-rinsed sterile plastic syringe and a pre-rinsed 0.45 μm
cellulose acetate filter (Sartorius) into polypropylene collection bottles. Water isotope samples were filtered streamside using
either 0.45 μm cellulose acetate (in 2019, 2020) or PES (in 2021) filters into 2mL glass (2019, 2020) or 25mL HDPE
scintillation (2021) vials filled with no headspace. Major ions and trace metals were sub-sampled from a 250 mL HDPE
collection bottle into 20 mL scintillation vials using rubber free syringes and a 0.45 μm PES filter. Water temperature and

specific conductance were measured onsite using a YSI EXO sonde. Bulk water samples were collected for microbial
community analyses in prepared HDPC bottles, radiocarbon ($\Delta^{14}$C-DOC) in Teflon bottles, and POC and $\delta^{13}$C-POC samples
in 4L HDPE plastic bottles. All bulk water samples for water chemistry and OC were filtered within 24 h of collection off site.
Samples for 16S rRNA gene sequencing were generally filtered within 4–12 h of collection, apart from the Bow River samples
that were filtered after 24 h of collection. To minimize the effects of differential processing times on assessing microbial

community dynamics, Bow River samples were excluded from our microbial analysis but retained in the hydrochemical and
carbon analyses.



Bulk water collected for 16S rRNA gene sequencing was processed using a 0.22 μm Sterivex (MilliporeSigma) filter and a peristaltic pump operated at 50–60 mL min$^{-1}$ to minimize cell breakage. Dissolved radiocarbon samples were filtered through

a 0.7 μm GF/F filter using a glass filter tower, collected into 1L amber glass bottles, and acidified to pH 2 using HPLC grade $H_3PO_4$. Material retained on the 0.7 μm GF/F filter was used for $\Delta^{14}$C-POC analysis. Samples for POC concentration and $\delta^{13}$C-POC were collected on 0.7 μm GF/F filters using plastic filter towers. Samples for DOC concentration and $\delta^{13}$C-DOC were acidified to pH 2 using trace metal grade HCl and cation and trace metal samples were acidified to pH 2 using trace metal grade $HNO_3$. Samples for DOC concentration, $\delta^{13}$C-DOC, $\Delta^{14}$C-DOC, DOM absorbance and fluorescence, anions, TDN, TDP and dSi were stored at 4°C; samples for $NH_4$, $NO_3+NO_2$, $\delta^{13}$C-POC, $\Delta^{14}$C-POC were stored at -20°C, and sterivex filters were


flash frozen in a liquid nitrogen dry shipper and upon return the laboratory stored at -80°C until analysis.

**2.3 Laboratory analyses**

**2.3.1 DOM absorbance and fluorescence, DOC, POC and OC isotopes**

DOM absorbance and fluorescence were analysed using a HORIBA Scientific Aqualog with a 1 cm path length quartz cuvette.
Absorbance scans were collected over a 240–800 nm wavelength range in 1 nm increments with a 0.5 s integration time. Excitation emission matrices (EEMs) were constructed over a 230–500 nm excitation wavelength range with an increment of 5 nm and a 5 s integration time, with an emission coverage increment of 2.33 nm with a 230–500 nm emission wavelength range.

DOC samples were analysed using a Shimadzu Total Organic Carbon analyser. Samples with less than 1 mg L$^{-1}$ DOC were analysed on a Shimadzu TOC-V equipped with a high sensitivity catalyst, using a 2 mL injection and five minute sparge time. Samples estimated (from absorbance) to have more than 1 mg L$^{-1}$ DOC were analysed using a Shimadzu TOC-L fitted with a regular sensitivity catalyst, using a 150 μL injection and five minute sparge time. For analyses on the TOC-V, a five point (0-1 mg L$^{-1}$) or six point (0-0.5 mg L$^{-1}$) calibration curve (R$^2$ > 0.98) was created daily using dilution from a 5 mg L$^{-1}$ stock
solution (SCP Science). For analyses on the TOC-L, a five point (0-1 mg L$^{-1}$ or 0-2 mg L$^{-1}$) calibration curve (R$^2$ > 0.98) was created through dilution of either 5 mg L$^{-1}$ stock solution (SCP Science) or a 10 mg L$^{-1}$ stock solution (SCP Science). Reference waters were created using dilution of a 5 mg L$^{-1}$ stock solution (SCP Science) or from a 1 mg C L$^{-1}$ caffeine solution. Reference waters and MilliQ blanks were run every ten samples and were within 10% of accepted values. Samples were blank corrected via the subtraction of mean blank concentrations of MilliQ samples run prior to ten sample groups to account for instrument
drift.



POC, $\delta^{13}$C-POC and $\Delta^{14}$C-POC samples were subjected to a heated acid fumigation following procedures outlined in Whiteside et al. (2011). Briefly, this involved heating the filters at 60°C for 24 h in a desiccator with 20 mL of concentrated trace metal grade HCl, then neutralizing the filters at room temperature for 24 h in a desiccator with NaOH pellets. Samples for POC

concentration and $\delta^{13}$C-POC were packaged into tin capsules before being measured using an Elementar VarioEL Cube Elemental Analyser a DeltaPlus Advantage isotope ratio mass spectrometer with a ConFlo III interface ($\delta^{13}$C-POC only) at the Environmental Isotope Laboratory at the University of Waterloo (in 2019) or the Jan Veizer Stable Isotope Laboratory (Ottawa, ON, Canada, in 2020 and 2021). $\delta^{13}$C-DOC samples were analysed using the wet oxidation method on an OI Analytical Aurora model 1030 wet TOC analyser connected to an Isotope Ratio Mass Spectrometer (IRMS) at the Jan Veizer Stable Isotope

Laboratory. $\Delta^{14}$C-POC samples were processed via organic combustion and $\Delta^{14}$C-DOC samples were processed using UV-oxidation, prior to graphitization and accelerator mass spectrometry analysis at the André E. Lalonde AMS Laboratory at the University of Ottawa (in 2019) or the National Ocean Sciences Accelerator Mass Spectrometry laboratory (NOSAMS; WHOI, Woods Hole, MA, USA, in 2020 and 2021; Batch numbers = 67580, 67380, 67069, 66828).

### 2.3.2 Hydrochemical samples

Nutrient ($NH_4^+$, $NO_3^-$+$NO_2^-$, TDN, TN, TP, dSi, SRP), trace metal (Al, Ba, Cr, Mn, Mo, Ni, Sr), and ion ($Mg^{2+}$, $Cl^-$, $Na^{2+}$, $SO_4^{2-}$, $Ca^{2+}$, $K^+$) samples were analysed at the Canadian Association for Laboratory Accreditation accredited Biogeochemical Analytical Service Laboratory at the University of Alberta (Edmonton, AB, Canada). Nutrient samples were analysed using flow injection analysis on a Lachat QuikChem 8500 FIA automated ion analyser and anions were analysed via ion chromatography on a Dionex DX600 . Cations and trace metals were analysed using Inductively Coupled Plasma-Mass

Spectrometry, on a Thermo Scientific iCAP Q (2019) and Agilent 7900 (2020-21). Water isotopes ($\delta^{18}$O-$H_2O$ and $\delta^2$H-$H_2O$) were analysed using a Picarro L2130 isotope and gas concentration analyser calibrated using standard references obtained from ice core water (USGS46) and Lake Louise drinking water (USGS47) (United States Geological Survey). Reference waters and MilliQ were run every 20 samples and were within 20% of accepted values. Sample values were calculated from an average of three injections where the standard deviation of $\delta^{18}$O-$H_2O$ was less than 0.2 and the standard deviation for $\delta^2$H-

$H_2O$ was less than 1; the first five injections were typically excluded due to memory effects. $\delta^{18}$O-$H_2O$ and $\delta^2$H-$H_2O$ were used to calculate deuterium excess, which we use as an indicator of water source given its known increase with elevation, and therefore water sourced from glaciers and high elevation snow (Bershaw et al., 2019; Bershaw et al., 2020; Boral et al., 2019)

### 2.3.3 Lab and bioinformatic processing of microbial samples

Bulk genomic DNA collected on the Sterivex filter was extracted using a DNeasy PowerWater Sterivex kit (Qiagen) according

to the manufacturer's instructions (Qiagen, 2019) with an amendment of a 1 h at 72°C for the initial incubation time, rather than 90°C for 5 mins, to increase DNA yield. Extracted samples were amplified using the 515F (5'GTGYCAGCMGCCGCGGTAA'3) and 926R (5'CCGYCAATTYMTTTRAGTTT'3) primers (Parada et al., 2016)



targeting the V4–V5 hypervariable region of the 16S rRNA gene with the following protocol: 3 min initial denaturation at 98°C, 35 cycles of: 30 s denaturation, 30 s primer annealing at ~ 60°C and 30 s extension at 72°C; and finally, 10 mins of final

extension at 72°C. Polymerase Chain Reaction (PCR) products were visualized on a 1.5% agarose gel and those samples showing product on the gel were subsequently purified using Nucleomag beads (ThermoFisher Scientific) at a 0.8 ratio of beads:sample. Unique indexes were then added to each sample using i7 and i5 adapters (Illumina) to construct the subsequent library. All amplicon and barcoded products from each respective year of sampling (2019, 2020, 2021) were verified on a 1.5% agarose gel and those that could be successfully visualized were then pooled (12.4 ng μL$^{-1}$, 20.17 ng μL$^{-1}$, 14.3 ng μL$^{-1}$).

The final quality of each pool was determined on an Agilent 2100 Bioanalyzer at the Molecular Biology Service Unit (Univ. of Alberta, Edmonton, AB, Canada) using a high sensitivity DNA assay prior to submitting the library for 16S rRNA gene sequencing. The final prepared 2–4 nM libraries, containing up to 50% PhiX Control v3 (Illumina, Canada Inc., NB, Canada) were sequenced on an Illumina MiSeq (Illumina Inc., CA, USA) using a 2 × 250 cycle MiSeq Reagent Kit v3 submitted at the Molecular Biological Services Unit (Univ. of Alberta) in 2019 and The Applied Genomics Core (Univ. of Alberta) in 2020

and 2021.

Sequence data was demultiplexed using MiSeq Reporter software (version 2.5.0.5) and Miseq Local Run Manager GenerateFastQ Analysis Module 3.0. The assembled data was then processed using the Quantitative Insights into Microbial Ecology (QIIME2) pipeline (Boylen et al., 2020, version 2021.11). Sequences were clustered into amplicon sequence variants

(ASVs) with chimeric sequences, singletons, and low abundance ASVs removed using DADA2 (Callahan et al., 2019). All representative sequences were classified with the SILVA v138 taxonomic database (Quast et al., 2013, Yilmaz et al., 2014), at the default 0.8 similarity cut-off. ASV sequences that were identified as eukaryotes or chloroplasts were removed. Samples were decontaminated by removing ASVs present in a sequenced field blank using the prevalence method with a threshold of 0.5 (Karstens et al., 2019; Parada et al., 2016b).


### 2.3.4 DOM absorbance and fluorescence calculations and PARAFAC analysis

EEMs were corrected for inner filter effects, Raman normalized, blank corrected and had Rayleigh and Raman scatter bands removed prior to parallel factor (PARAFAC) analysis (Murphy et al., 2013). PARAFAC analysis was performed in Matlab (Version 9.12.0) using the drEEM toolbox (Version 0.6.5) (Murphy et al., 2013). PARAFAC models of up to seven

components were assessed and samples with high leverage were removed. A four-component model was validated using split-half analysis and was ultimately selected based on a low sum of square error and visual confirmation that only random noise remained in the residuals. Absorbance scans were utilized to calculate specific UV absorbance at 254 nm (SUVA$_{254}$; a measure of DOM aromaticity) (Weishaar et al., 2003) and the spectral slope coefficient between 275 and 295 nm (S$_{275-295}$; a measure of DOM molecular weight) (Helms et al., 2008). Corrected fluorescence data were utilized to calculate the fluorescence index





(FI) (McKnight et al., 2001b), humification index (HIX) and biological index (BIX) (Huguet et al., 2008), and the proportional contribution of common fluorescence peaks (peaks A, B, C, M, T; (Coble, 1996)). Further description of absorbance and fluorescence-based metrics is provided in Table S2.

## 2.4 Data analysis

### 2.4.1 Organic matter

Three way ANOVAs followed by Tukey's honest significant difference (HSD) tests were performed to determine whether OC concentration, OM character, and OC isotopic composition varied across distance bins, hydrological periods, and years. Linear regressions were performed to assess the relationship between deuterium excess and PARAFAC components, and the relationship between $\Delta^{14}$C-DOC and PARAFAC components. We use deuterium excess as proxy for glacial meltwater because it is known to increase with elevation (Bershaw et al., 2019) and is thus typically higher in glacial ice (Souchez et al., 1983)

and water contributions from high elevation snow (Bershaw et al., 2020) when compared to lower elevation (downstream) water sources. To explore how DOM parameters varied across hydrologic seasons and with distance downstream a principal component analysis (PCA) was performed. Deuterium excess was fitted onto the ordinated PCA as a passive overlay (using envfit; (Oksanen, 2007)).

### 2.4.2 Microbial analysis

To enable contrasts between sites most strongly influenced by glaciers (i.e., the headwater and near sites) and those further downstream, the mid distance range sites were excluded from statistical analysis of the microbial samples. Alpha diversity was calculated using the Shannon index (Hill, 1973) and beta diversity was assessed using non-parametric multi-dimensional scaling (NMDS) with a Bray Curtis distance matrix created from Hellinger-transformed ASV abundance data (Ramette, 2007; Legendre and Gallagher, 2001). Significant differences between clusters on the NMDS were assessed using permutational

multivariate ANOVAs (perMANOVA). A backward step redundancy analysis (RDA) was also performed on Hellinger-transformed ASV abundance data and non-correlated normally scaled environmental parameters. Initially considered environmental parameters in our RDA included temperature, pH, specific conductance, turbidity, DOC concentration and DOM composition (% humic-like [see Section 3.1.2], FI, BIX, HIX, peak A, peak B, peak C, peak M, peak T, $S_{275-295}$, SUVA$_{254}$), nutrients (TP, TDN, dSi), major ions (Cl, SO$_4$, Ca, K, Mg, Na), and trace metals (Al, Ba, Cr, Mn, Mo, Ni, Sr). Of

this originally-considered suite, we omitted highly correlated parameters to avoid an over-parameterized model [Section 3.2]. An indicator species analysis (ISA) was performed on raw (non-transformed) ASV abundance data (Dufrêne and Legendre, 1997). Co-variation between indicator species and environmental parameters (nutrients, cations, anions, OC, glacial melt input, discharge, water temperature, and specific conductance) was assessed using Spearman's rank correlation.





All data was analysed with the R programming language, using *vegan* (Oksanen, 2007), *stardom* (Pucher et al., 2019), *decontam* (Davis et al., 2018), *indicspecies* (Cáceres and Legendre, 2009), and base packages (R Core Team, 2022). Data visualization utilized the package *ggplot2* (Wickham, 2016).

## 3 Results

### 3.1 Spatial and temporal variation in stream organic matter characteristics

**3.1.1 Water isotopes, DOC concentration and DOM composition**

Across all sampling years (2019–2021), deuterium excess was higher at the headwater and near sites relative to mid and far sites; seasonally, deuterium excess was greater during melt and post-melt periods relative to pre-melt periods ($p < 0.0001$, Fig. S1, Table S3). $\delta^{18}O\text{-}H_2O$ became more enriched from pre-melt, to melt, to post melt seasons ($p < 0.0001$, Table S4), and during pre-melt, was enriched at far sites relative to near sites (Fig. S1).


Across all seasons and years, DOC concentrations were universally low closest to the glacier (0.09–0.76 mg L$^{-1}$) and increased downstream (i.e., the far sites had significantly higher DOC concentrations compared to the headwater, mid and near sites, and the mid sites had significantly higher concentrations than the near sites; $p < 0.001$) (Fig. 2a, Table S5). Between seasons, pre-melt DOC concentrations were higher than during the melt season in 2019 only, and between years, concentrations during 295    2019 were greater than in 2020 and 2021 ($p < 0.01$) (Fig. 2a, Table S5). In comparison, POC concentrations were highest at headwater sites, with concentrations dropping significantly between headwater and near sites, and then showing a gradual, but largely non-significant trend towards increasing concentrations with further distance downstream (Fig. 2d, Table S6). Across all sites and dates, POC concentrations ranged from 0.04 mg L$^{-1}$ to 2.8 mg L$^{-1}$, and similar to DOC, were greater in 2019 relative to 2021 (Fig. 2d, Table S6) ($p < 0.05$). POC was typically the dominant component of the total OC (TOC) pool at 300    headwater sites (POC:TOC = 0.32–0.89; median value > 0.5 for all but pre-melt 2021), and showed a relative decline with distance from the glacier (POC:TOC = 0.1–0.52 at the furthest downstream site) (Fig. 2e).

From the suite of PARAFAC models considered, a four component model was selected (Fig. S2). Using the OpenFluor database (Murphy et al., 2014), components 1 (C1) and 2 (C2) were identified as humic-like components, component 3 (C3) 305    was identified as a tryptophan-like (protein-like) component, and component 4 (C4) was identified as a tyrosine-like (protein-like) component (Table S7). The proportion of protein- (sum of C3 and C4) and humic- (sum of C1 and C2) like components varied across our distance bins (headwater, near, mid and far), with the proportion of protein-like DOM decreasing with increasing distance downstream, and the proportion of humic-like DOM showing the inverse relationship ($p < 0.05$) (Fig. 2b–c, Table S8). Broadly speaking, the proportion of protein-like DOM increased from pre-melt to melt periods, while humic-like





DOM decreased, with this trend being most prominent at mid and far sites ($p = 0.04$) (Fig. 2b–c, Table S8). We observed modest relationships between deuterium excess and both DOC concentration and DOM composition, with DOC concentration decreasing with increasing deuterium excess, and the proportion of protein-like DOM increasing ($R^2_{adj} = 0.28$, $p < 0.0001$; $R^2_{adj} = 0.23$, $p < 0.0001$) (Fig. 3).

### 3.1.3 Particulate and dissolved OC isotopes

$\delta^{13}$C-DOC ranged from -35 to -20.2‰ (Figs. 4 and S3) across sites and was significantly more depleted during the pre-melt and melt seasons relative to the post-melt season, and in 2020 and 2021 compared to 2019 (Table S9). $\delta^{13}$C-POC ranged from -38.8 to -20.8‰ (Fig. 4) and across all sites was significantly more depleted during the pre-melt season ($p < 0.001$) and in 2019 ($p < 0.001$) (Table S10). Spatially, $\Delta^{14}$C-DOC generally became more depleted with increasing proximity to glacier termini ($p=0.08$, Table S11) but exhibited high variability at the headwater site during the melt season (Figs. 4 and 5a). $\Delta^{14}$C-

DOC values ranged from -554‰ to +7‰ ($n = 8$) at the Athabasca glacier headwaters, -634‰ to +18‰ at near sites ($n = 16$), -404‰ to +27‰ at mid sites ($n = 24$) and increased to +71‰ during the pre-melt season ~50km downstream from glacier terminus ($n = 1$) (Fig. 5). $\Delta^{14}$C-DOC also varied seasonally, with pre-melt and melt season stream water being more enriched in $\Delta^{14}$C-DOC relative to post-melt stream water ($p = 0.003$) (Fig. 5a, Table S11). In addition, $\Delta^{14}$C-DOC showed a modest negative correlation with protein-like DOM ($R^2_{adj} = 0.15$, $p = 0.006$) (Fig. 5b). Similar to $\Delta^{14}$C-DOC, $\Delta^{14}$C-POC exhibited a

large range in values, from -550‰ to -108‰ (Fig. 4). However, in contrast to $\Delta^{14}$C-DOC the large variation in $\Delta^{14}$C-POC values was not associated with seasonal or spatial differences between samples (Fig. 4).

### 3.1.4 An integrative assessment of DOM composition across sites

A principal component analysis exploring DOM composition across sites identified eight principal components with 64% of the observed variation being explained by the first two principal component axes (Fig. 6). Principal component 1 (PC1)

explained 48% of sample variance, with humic-like PARAFAC components, DOC concentration and HIX (humification) loading positively on this axis, and protein-like PARAFAC components, BIX (biological origin), FI (microbial origin), and $S_{275-295}$ (declining molecular weight) loading negatively. Sites generally plotted along PC1 according to their distance range, with headwater and near sites more frequently having negative PC1 scores, and far sites having more positive PC1 scores (Fig. 6). PC2 explained 16% of sample variance and was positively associated with $\delta^{13}$C-DOC and negatively associated with

SUVA$_{254}$ (aromaticity). A passive overlay of deuterium excess was negatively associated with PC1, and positively associated with PC2 (Fig. 6).

### 3.2 Microbial community characterization and its controls over space and time

Generally, microbial communities shared similar dominant phyla across sites and seasons with the top ten most abundant phyla across all samples being: Acidobacteriota, Actinobacteriota, Bacteroidota, Bdellovibrionota, Chloroflexi, Cyanobacteria,





Firmicutes, Plantomycetota, Proteobacteria, and Verrucomicrobiota (Fig. S4). At near and far sites in the Sunwapta-Athabasca and North Saskatchewan rivers, these ten phyla collectively described between 80–88% of the resolved microbial community composition (Fig. S4). To explore taxonomic variation at a finer scale, we investigated shared ASVs across distance ranges. This assessment revealed a core microbial community, shared among all our samples, comprised of 1409 ASVs total, with the top 10 most abundant taxa belonging to families Comamonadaceae, Cyanobiaceae, Gallionellaceae, Ilumatobacteraceae,

Sporichthyaceae, Alcaligenaceae, Methylophilaceae and Flavobacteriaceae. While this core community only represented 3.4% of all identified ASVs (Fig. 7a), it made up a large fraction of relative abundance at each site (between 25–80%; Fig. 7b), with a general decreased prevalence of the core community at the far sites, when compared to headwaters and near. Accordingly, microbial community diversity was highest at the far sites during all seasons (Fig. S5), and a comparison of microbial diversity between samples (beta diversity) showed that community composition shifted significantly with movement from headwater

and near to far sites ($R^2 = 0.09$, $p < 0.001$) (Fig. 8, Table S12). Communities were also distinct across rivers ($R^2 = 0.08$, $p < 0.001$; note that the Bow was excluded from microbial analyses, see section 2.2) and different sampling years ($R^2 = 0.06$, $p < 0.001$) with each river and year being significantly different from one other ($p<0.01$) (Fig. S6, Table S12).

We used RDA to assess whether environmental factors could explain microbial community variance. To avoid an over-

parameterized model, we omitted highly correlated environmental variables from initial consideration in the model, reducing the initially-considered factors described in the Methods to: % protein-like DOM, water temperature, DOC concentration, POC concentration, $\delta^{13}$C-DOC, $\delta^{18}$O-H$_2$O, deuterium excess, S$_{275-295}$, SUVA$_{254}$, TN, TP, and specific conductance. In total, the RDA described a 9.1% (adjusted) shift in microbial community structure, with % protein-like DOM, water temperature, specific conductance, POC and deuterium excess identified as significant predictor variables ($p < 0.001$) (Fig. 9). RDA axis 1 (RDA1)

described a gradient of apparent high elevation (e.g., glacial ice, snow) water source inputs, with deuterium excess and protein-like DOM negatively correlated, and specific conductance and water temperature positively correlated with this axis (Fig. 9). RDA axis 2 (RDA2) showed water temperature loading negatively on this axis and POC loading positively (Fig. 9). Headwater sites plotted negatively on RDA1 and positively on RDA2, whereas far sites tended to plot positively on RDA1. Overall, headwater microbial communities were associated with relatively higher proportions of protein-like DOM, deuterium excess,

and POC but lower specific conductance and water temperature compared to communities at the far sites (Fig. 9).

### 3.2.1 Indicator species analysis

An ISA resolved three strong (indicator value (IV) > 0.8, $p < 0.05$) indicator species for headwater sites and two strong indicator species for far sites. There were no strong indicator species for near sites at this IV threshold. Indicator species for the headwater sites were identified as a species from the genus *Cryobacterium* (IV = 0.89), a species from family Beggiatoaceae

(IV = 0.82), and a species from family Microbacteriaceae (IV = 0.87). Far indicators were a species from family Sporichthyaceae (IV = 0.83), and a species in the genus *Cyanobium_PCC-6307* (IV = 0.83). Correlations of indicator ASVs



with deuterium excess revealed that the relative abundance of indicator species was strongly positively (headwater sites) and negatively (far sites) correlated with deuterium excess (Fig. 10).

The strong indicator species (IV > 0.8) that were identified at the headwater sites were correlated ($p < 0.05$) with various stream characteristics (Fig. 11). All headwater indicator species were negatively correlated with water temperature, while *Cryobacterium sp*. and Microbacteriaceae *sp*. were also negatively correlated with $Cl^-$, $K^+$, $Mg^{2+}$, $Na^+$, TN, dSi, and humic-like DOM, and positively correlated with protein-like DOM and deuterium excess (Fig. 11). Far site site indicator species were positively correlated with dSi, $SO_4^{2-}$ and $Na^+$ (Fig. 11).

**4 Discussion**

**4.1 Changing organic matter quantity and character driven by increased soil development downstream**

Stream OM originates from two primary sources: 1) in situ primary production of OM from autotrophic microbes or macrophytes (autochthonous carbon); or 2) OM transported into the stream system from outside sources (allochthonous carbon) such as nearby vegetation and soils (Hinton et al., 1997, 1998; Tanentzap et al., 2014; Table S13). In glacially-fed

streams, glacially sourced OM also contributes to the allochthonous carbon pool (Hood et al., 2015a). However, glaciers are associated with a series of variable OM sources, as a result of in situ microbial metabolism on the glacier surface (supraglacial) and at the glacier bed (subglacial) (Hotaling et al., 2017; Stibal et al., 2012), as well as from windblown deposition on the snow and ice surface and comminuted sediments and bedrock at the glacier bed (Hood et al., 2020; Stubbins et al., 2012). Thus, the changing water source contributions to streams is a critical determinant of fluvial OM composition, both in alpine

environments and elsewhere (Fasching et al., 2016).

This study identified a shift in OC quantity and OM character with increasing distance downstream from glaciers and thus declining relative contributions of glacial meltwater to streamflow. Low DOC concentrations in our mountain headwater streams, which drain catchments lacking developed soils and where dilute glacial meltwaters are the primary source of

allochthonous carbon, are in agreement with previous work from the Alps (Singer et al., 2012), Tibetan Plateau (Zhang et al., 2018; Zhou et al., 2019), Canadian Rockies (Lafrenière et al., 2004) and the Greenland Ice sheet (Kellerman et al., 2020) . In contrast, the large relative contribution of POC to the headwater TOC pool is consistent with increased erosional processes in high slope glacial margin environments, export of glacially derived sediments (Hood et al., 2020) and higher rates of particle re-suspension in these high gradient, turbulent, reaches (Marcus et al., 1992). At the headwaters, the protein-like character of

the DOM pool suggests dominant contributions from microbial production or reworking of DOM, either on the glacier surface or at the bed, as has been found in many other glacierized regions (Zhou et al., 2019; Dubnick et al., 2010; Hood et al., 2009; Stubbins et al., 2012; Pain et al., 2020; Singer et al., 2012). Increasing DOC concentrations and a shift from protein-like to



humic-like DOM with increasing distance downstream supports increasing contributions of allochthonous carbon from surrounding vegetation and more developed soils (see also, Zhou et al., 2019). The conclusion that headwater sites receive a
majority of their OM directly from the glacier, and downstream locations gain additional OM contributions derived from catchment soils, is supported by declines in deuterium excess with movement downstream (Fig. S1), and thus an inferred switch to increasing water contributions of rainfall and lower-elevation sourced precipitation (Bershaw and Lechler, 2019; Bershaw et al., 2020).

**4.2 Stream POC sourced from plant carbon with potential additions from glacier microbes**

Compared to the DOC pool, the POC pool at the headwater, near and mid distance sites displayed less overall isotopic variation, pointing to a more consistent POC source to streams. Across distance ranges and seasons, the POC pool was consistently aged ($\Delta^{14}$C-POC = -550‰ to -108‰). At the headwaters, this depletion likely indicates sourcing from glacial export of overridden vegetation (Bhatia et al., 2013), or fossil fuel products (Stubbins et al., 2012) and the heterotopic consumption of fossil fuels by cryoconite microbes (Margesin et al., 2002). Further downstream, radiocarbon depleted POC could be sourced from the
addition of water from aged soil margins, or glacially exported aged POM that has persisted through the stream network (Hood et al., 2020). However, save for the headwater site during the pre-melt period, $\delta^{13}$C-POC indicated POM primarily of terrestrial origin ($\delta^{13}$C‰ = -26 to -28‰), with occasional slight depletions at headwater and near sites ($\delta^{13}$C‰ = -29 to -30‰) pointing to possible contribution from subglacial chemosynthetic biofilms (Table S13). Indeed, if some of the aged POM is sourced from subglacial microbes, this fraction of POM may be more accessible to downstream food webs than aged terrestrial POM
(Brett et al., 2017). The pre-melt headwater sample was uniquely $\delta^{13}$C-enriched compared to other POM samples, which may suggest microbial re-working of depleted C sources as a more dominant POM source at this time.

**4.3 Seasonal patterns in DOM from headwaters to downstream reaches**

*Seasonal consistency at headwater sites*

Over the past two decades, studies from polar and alpine regions across the globe have led to the characterization of glacial OM as universally dilute, of microbial origin and yet, paradoxically, old (i.e. radiocarbon depleted) (Bhatia et al., 2013; Hood et al., 2009). Seasonally, we found that the character of the OM comprising our headwater DOM pool was consistently protein-like across the pre-melt, melt, and post-melt seasons, with little variation. This contrasts with other studies where melt-season shifts in glacial stream DOM character from relatively more humic-like to more protein-like have been associated with
changing drainage flow paths accessing different OM sources (Kellerman et al., 2020). For example, DOM fluorescence in the outflow draining Leverett Glacier, a large land-terminating glacier of the Greenland Ice Sheet, was relatively more humic-like in the early season, when waters draining subglacial flow paths access terrestrial material from previously overridden soils and bedrock dominated stream contributions (Kellerman et al., 2020). In comparison, during peak melt flow, when supraglacial inputs dominated, Leverett Glacier outflow DOM had a distinctly more protein-like fluorescence signature (Kellerman et al.,



2020). The observed consistency in headwater DOM character observed in this study may suggest that, regardless of flow path, protein-like OM pools are always accessible at these relatively smaller Rocky Mountain glaciers. Water at the Athabasca glacier terminus during the pre-melt period is predominately sourced from the distributed subglacial drainage network (Arendt, 2015), with the DOC pool being aged ($\Delta^{14}$C-DOC = -175‰) and $\delta^{13}$C-DOC depleted (-25.5‰, Fig. 4). These carbon isotopic signatures, in conjunction with the consistent protein-like DOM fluorescence, suggest microbial reworking of previously

overridden vegetation and soils at the glacier bed (Bhatia et al., 2013; Dubnick et al., 2010; see also Table S13). During the melt season, headwater $\delta^{13}$C-DOC shifts towards an autotrophic signature, likely sourced from chemosynthetic ($\delta^{13}$C-DOC = -29‰ to -35‰) or photoautotrophic ($\delta^{13}$C-DOC = -25‰ to -20‰) microbial communities, but the $\Delta^{14}$C-DOC signature generally remains depleted. Taken together, these compositional metrics suggest either: a) chemolithoautotrophy tied to microbial heterotrophic consumption of ancient vegetation and soil at the glacier bed (Hood et al., 2009); or b) microbial

incorporation of radiocarbon-dead fossil fuel combustion products deposited on the glacier surface (McCrimmon et al., 2018). Previous work in alpine glacial systems in both Alaska (Stubbins et al., 2012) and Tibet (Spencer et al., 2014) have found the latter to substantially contribute to supraglacial DOC pools during peak melt periods. In some melt season samples, however, headwater $\Delta^{14}$C-DOC was enriched (Fig. 4). This could be indicative of contributions from recent photosynthetic supraglacial microbial communities (Kellerman et al., 2021; Smith et al., 2017; Stibal et al., 2012) or wildfire-derived soot present on the

glacier surface (Aubry-Wake et al., 2022; Nizam et al., 2020).

*Seasonal variation downstream*

In contrast to the headwaters, downstream sites displayed clear seasonal variation in OM character. During the pre-melt period, inputs of enriched $\Delta^{14}$C-DOC were coupled with a $\delta^{13}$C-DOC signature consistent with terrestrial OM inputs (-25‰ to -27‰)

at the near and mid sites, likely indicative of snowpack melt draining surficial soils, leading to pulses of concentrated modern, terrestrial-origin DOM delivery to streams (Fig. 4). This flushing of concentrated DOM from soils was the likely cause of the spike in DOC concentration observed during the pre-melt season at downstream sites in 2019 (Fig. 2), when sampling captured the spring freshet. This process has been observed previously at Bow River from 1998–2000, when the highest DOC concentrations were found to occur during the initial snowpack melt and, as in this study, were followed by a decline in river

DOC concentrations (Lafrenière and Sharp, 2004). These isotopic signatures were also tied to higher proportional inputs of humic-like DOM (Fig. 4).

During the melt season, a proportional increase in protein-like fluorescence at the near, mid, and far sites was coupled with DOM pools at near and mid sites switching from being consistent with allochthonous terrestrial inputs ($\delta^{13}$C-DOC = -25 ‰ to

-29 ‰) to showing a clear autochthonous signature reflective of chemo- or photosynthetic OM production ($\delta^{13}$C-DOC = -29‰ to -35‰ and $\delta^{13}$C-DOC = -25‰ to -20‰; Fig. 4 and S2). Near and mid sites were also characterized by DOM that was often $\Delta^{14}$C-DOC depleted. This trend may reflect increasing in situ production by microbial photoautotrophs, as evidenced by the presence of cyanobacteria DNA in the microbial amplicon libraries from these sites (Fig. S4), or microbial re-working of





humic-like DOM, either within the stream or in catchment soils (Fig. 4). Aged carbon downstream likely represents the mixing

of either: a) ancient glacially exported material from the headwaters persisting downstream or b) additions from deeper, older soils (Shi et al., 2020). During the post-melt period, stream water at the near and mid distances was either aged ($\Delta^{14}$C-DOC = -630‰, -400‰) with a $\delta^{13}$C-DOC terrestrial signature (-28‰, -29‰), consistent with OM contributions from deeper soils when discharge is declining, or slightly more modern ($\Delta^{14}$C-DOC = -193‰, -110‰) and $\delta^{13}$C-enriched (-22‰), consistent with incorporation of autochthonous primary production into the DOM pool (Fig. 4).

**4.4 Implications of inferred composition for OM lability**

The transition from recently produced, low molecular weight, protein-like DOM to high molecular weight, humic-like DOM with movement downstream has implications for DOM lability. Protein-like DOM is associated with free amino acids and proteinaceous compounds which are generally considered to be more available for heterotrophic microbial consumption (Coble, 1996), while factors such as decreasing molecular weight ($S_{275-295}$) (Fig. 6) are also associated with increasing OM

lability (Moran and Zepp, 1997; Patriarca et al., 2021). In previous studies of glacial DOM, paired high resolution mass spectrometry and UV-Vis spectrometry have shown that protein-like fluorescence is associated with unsaturated aliphatic compounds, identified as being peptide- and lipid-like (Kellerman et al., 2020, 2021; Stubbins et al., 2012; Zhou et al., 2019), which have been found to be a common contributor to glacier DOM pools in many regions (Singer et al., 2012; Bhatia et al., 2013; Kellerman et al., 2021; Stubbins et al., 2012; Zhou et al., 2019). In contrast, humic-like DOM is associated with complex

material (e.g., increased lignin content; Mann et al., 2016; Zhou et al., 2019) and thus is generally considered to be a more recalcitrant fraction of DOM. This shift in lability has important implications for stream ecosystems, because microbial consumption of DOM can provide an energetic transfer to higher trophic levels (e.g., Fellman et al., 2015), with labile DOM being linked to increased microbial productivity (Pontiller et al., 2020). Labile OM released at the headwater site could also initiate priming of the OM pool, resulting in increased microbial consumption of recalcitrant soil-derived OM (Bingeman et

al., 1953; Guenet et al., 2010). Notably, the importance of priming has yet to be assessed in glacially fed streams (Bengtsson et al., 2018). However, despite the shift in apparent bulk DOM lability with movement downstream, one additional consideration is the low overall DOC concentration at headwater sites. Although downstream sites had smaller proportions of protein-like DOM, these sites had higher DOC concentrations overall. A rough calculation to compare between sites yields an estimated mean 0.28 mg L$^{-1}$ of protein-like DOM at downstream sites, compared to ~0.21 mg L$^{-1}$ at the headwaters (using the

mean DOC and % protein-like DOM for each distance bin). Therefore, although DOM may have been more accessible to microbes on a proportional basis in the headwaters, overall OM processing may still have been higher downstream.

**4.5 Microbial community structured by environmental controls and dispersal**

Microbial communities and OM pools are inherently linked since microbes contribute both to the creation of OM and its consumption (Kujawinski, 2011). To further explore this relationship along our river transects, we investigated how microbial





community composition shifts with increasing distance from glaciers and coupled changes in OM source, OM character, and
      other physicochemical characteristics. Across all sites and seasons, 10 phyla were found to account for 80–88% of community
      composition (Fig. S4). While some of these phyla (Actinobacteria, Acidobacteriota, Cyanobacteria, Bacteroidota, and
      Proteobacteria) are ubiquitous in freshwater (Tamames et al., 2010), many (Proteobacteria, Bacteroidota, Acidobacteriota,
      Actinobacteriota, Chloroflexi, Cyanobacteria and Firmicutes, Plantomycetota) are also commonly identified as being dominant
in glacial ecosystems and in glacially-fed streams (Boetius et al., 2015; Bourquin et al., 2022; Hotaling et al., 2019). These
      glacially associated phyla are known to include taxa that are adapted to the environmental conditions typical of glacial
      environments, such as oligotrophy and cold temperatures (Bourquin et al., 2022).

*Environmental controls and species sorting*

At the ASV level, several lines of evidence indicate that glacial inputs and changing environmental conditions with movement
      downstream have an effect on microbial community structure. First, within site (alpha) and between site (beta) diversity
      increased with increasing distance from the glacier (Fig. 7, Fig. S5). This result is consistent with a previous study in the Alps,
      where lower alpha diversity near glaciers was attributed to a decrease in the diversity of microbial source pools at higher
      elevations (e.g., lower groundwater contributions) and/or harsher upstream environmental conditions (Wilhelm et al., 2013).
Second, we identified specific microbial indicator species for both the headwater and downstream sites. Headwater indicators
      Microbacteriaceae *sp.* and Microbacteriaceae *Cryobacterium sp.* are heterotrophs that are commonly found within glacial
      cryoconite holes and are believed to possess specific adaptations for cold temperatures and nutrient-poor conditions (Liu et al.,
      2020). The third headwater microbial indicator was a sulfur oxidizing chemolithotroph (Beggiatoaceae *sp.*), likely sourced
      from the subglacial environment where chemosynthesis is common (Anesio et al., 2017). The identification of these indicator
species suggests that unique microbes (sourced from both the supraglacial and subglacial environment) may be seeding the
      fluvial microbial community at the glacial headwaters (Wilhelm et al. 2013; Bhatia et al., 2006). Third, we found various lines
      of evidence supporting some species sorting of microbial taxa as a result of local environmental conditions (Lagenheder et al.,
      2011; Van der Gucht et al., 2007). Redundancy analysis identified a series of physiochemical variables – including protein-
      like DOM, water temperature, POC concentration, and specific conductance – as significant drivers of microbial community
composition across sites (Fig. 9). Deuterium excess, an indicator of high-elevation water source, was also a significant
      descriptor of microbial community structure. In addition, the heterotrophic headwater indicators, in particular, were negatively
      correlated with nutrients and ions (dSi, TN, $Ca^{2+}$, $K^+$, $Mg^{2+}$, $Na^+$) and warmer temperatures, and positively correlated with
      protein-like DOM, while downstream microbial indicators generally showed the opposite correlations. Of these environmental
      controls, DOM character can select for different specialized heterotrophic microbes (Judd et al., 2006), water temperature
broadly controls rates of microbial activity  (D'Amico et al., 2006), and POC can shape bacterial extracellular enzyme
      requirements and thus, microbial community composition (Kellogg and Deming, 2014). Thus, glacial meltwater contributions
      can impact microbial community structure both by altering stream abiotic factors (e.g., water temperature, conductivity,
      nutrients, carbon (Milner et al., 2017)) and by being a biotic source of novel microbes (Wilhelm et al., 2013; Bhatia et al.,



2006). Overall, it appears that environmental controls, including DOM character, play some role in structuring microbial
community composition in these glacially fed streams.

*Dispersal*

Despite this evidence for glacial seeding of microbial communities and presence of significant environmental drivers of
microbial community composition, we note that the RDA explained only a small amount (9%) of the observed variation in
community structure (Fig. 9). Similarly, perMANOVA tests of species divergence among site bins (i.e., beta diversity)
indicated that spatial variation only explained a small proportion of community composition, suggesting a high degree of
similarity between microbial communities ($R^2$ = 0.09, p < 0.001, Table S12). This low overall explanatory power of specific
environmental controls, combined with the observed community divergence between environmentally disparate sites, is
consistent with a recent synthesis effort that concluded environmental drivers are not always good predictors for microbial
community composition in streams (Zeglin, 2015).

This lack of strong environmental control is also consistent with patterns observed for the "core" group of microbial ASVs
that was ubiquitous across all distance ranges (headwater, near, far) examined in this study. While this core ASV group
accounted for only ~3% of the total identified ASVs, it represented a large proportion of average relative ASV abundance
across sites, ranging from a median of 70% of the total community relative abundance at headwater locations to 55% at
downstream sites (Fig. 8). Across all sites, the most abundant (top 10) microbial taxa from these core taxa display diverse
metabolisms, with some heterotopic taxa having species that are considered to be generalists (e.g., Flavobacterium (Zheng et
al., 2019), Sporichthyaceae), specialists for methanol consumption (Methylophilaceae (Beck et al., 2014)), specialists for
consumption of terrestrial carbon (Alcaligenaceae *sp.*) and autotrophic taxa, including both photoautotrophs (e.g.,
cyanobacteria) and chemoautotrophs (e.g., iron oxidizing Gallionellaceae (Hallbeck and Pedersen, 2014)). Increases in alpha
diversity and declining relative importance of the "core" community from headwaters to downstream sites indicates more
diverse communities downstream, consistent with an increased variety of sources of microbial communities from the
surrounding terrestrial environment (Fig. S5). However, the strong persistence of the core community across our transects also
reinforces that rivers are highly connected environments that can facilitate high rates of dispersal (Tonkin et al., 2018). As
such, mass effects can be strong drivers of microbial community composition in fluvial networks resulting in community
homogenization and, overall, decreased species sorting (Crump et al., 2007; Evans et al., 2017; Leibold et al., 2004; Pandit et
al., 2009). These findings contrast with those from studies of more isolated lakes (Logue et al., 2010; Van der Gucht et al.,
2008) and soils (Fierer et al., 2006), where environmental divergence between sites is better able to drive divergence in
microbial community composition. In our glacially-fed systems, particularly, we find that while local environmental conditions
exert some control over microbial community composition, mass effects are likely more influential, suggesting a clear
ecological connection from headwaters to 100-km downstream sites, despite striking environmental change.



## 5 Conclusions

This study illustrates that fluvial OM quantity and quality changes along stream transects from glacier headwaters to ~100 km downstream in the Rocky Mountains. These changes occur as a result of shifting inputs from different OM sources, with
headwater sites adjacent to the glacier containing DOM that is compositionally consistent with predominantly autotrophic inputs (chemosynthesis and photosynthesis) and microbial reworking of terrestrially sourced OM that is aged and likely labile. Comparatively, the downstream DOM pool was predominately humic-like, but exhibited seasonal shifts in apparent OM source with some apparent allochthonous inputs in the pre-glacial melt and post-glacial melt seasons, and autochthonous sources increasing at the height of the melt season. This study also identified that a portion of glacially exported POC may be sourced
from microbial activity, and thus is potentially accessible to downstream food webs. In contrast to these clear environmental gradients, we found that, despite evidence for glacial seeding of headwater microbial communities, mass effects appeared to enable dispersal of a persistent core microbial group throughout our stream transects from the glacial headwaters to ~100 km downstream.

Overall, findings from this work will enable better predictions about the impacts of glacial retreat on stream ecosystems, and the loss of labile OM inputs in Alpine headwater regions. In the future, with increased glacier retreat in the Canadian Rockies, glacially-fed rivers will shift from a glacier ice melt regime towards one dominated by seasonal snow and rainwater inputs (Pradhananga and Pomeroy, 2022). Necessarily, these changes will be accompanied by increases in surrounding soil development and vegetation cover. The likely result is a higher proportion of humic-like DOM at headwater sites, a potential
loss of glacially-derived POM bioavailable to downstream primary consumer organisms (Cotner and Biddanda, 2002), and changes in other environmental parameters (temperature, chemistry) identified in this study as being important for structuring stream microbial communities. At the same time, a loss of glacially exported microbes will alter the seed pool at headwater sites, which appears to persist far downstream, thus potentially altering the core community that is characteristic of this glacially-fed river system. Overall, glacier retreat and eventual deglaciation appears poised to cause a loss of microbial
biodiversity in the Canadian Rockies, similar to predictions for other glacierized Alpine regions (Hotaling et al., 2017). This biodiversity loss at the base of stream food webs could have a negative impact on overall riverine ecological stability and downstream ecosystem services via the loss of unique metabolic functions.




**Data availability**

All water chemistry and organic matter datasets generated and analysed in this study are available on PANGEA repository (Felden et al., 2023)(PDI-35664 and Serbu et al., submitted). Microbial 16S rRNA gene sequences generated and used in this study can be accessed from the NCBI database, using accession number PRJNA995204.

**Author contributions**

VSL, JAS, SET, and MPB designed the field program, and SET and MPB designed this study. HD and JAS led the field sampling campaigns, and HD, MC, and JAS conducted the laboratory analysis. MC built the 16S rRNA gene libraries and conducted the bioinformatic analyses. HD conducted all other data and statistical analyses, and visualized the results. HD wrote the paper with input from SET and MPB, and all authors provided comments on the manuscript.

**Competing interests**

The authors declare that they have no conflict of interest.

**Acknowledgements**

Data collected via fieldwork was approved under permits JNP-2018-29597 (Jasper National Park) and LL-2019-32266 (Banff National Park). We are grateful to L. Thomas, S. Metacat-Yah, R. Buford, P. White, S. Enns and J. Flett for invaluable assistance in the field. L. Thomas and S. Metacat-Yah were funded by the University of Alberta I-STEAM program and an NSERC USRA award (to L. Thomas). R. Buford was funded by a MITACS Globalink Research Award. We would like to thank the Biogeochemical Analytical Service Laboratory (BASL; University of Alberta) and The Canadian Centre for Isotopic Microanalysis (CCIM; University of Alberta) laboratory managers and staff for their role in the analysis of our samples. This work was funded by a Canadian Mountain Network grant to V. St. Louis, S. Tank, and M. Bhatia; Campus Alberta Innovates Program funding to M. Bhatia and S. Tank; and an Alberta Conservation Association Grant in Biodiversity to H. Drapeau. H. Drapeau was funded in part by an NSERC CGS-M postgraduate fellowship. We acknowledge that the University of Alberta is on Treaty 6 territory and fieldwork was conducted on Treaty 7 and 8 territories. Jasper and Banff National Parks are located on the lands of the Ktunaxa ʔamakʔis (Ktunaxa)**,** As'in'i'wa'chi Niy'yaw Askiy (Rocky Mountain Cree), Îyãħé Nakón mąkóce (Stoney), Niitsítpiis-stahkoii ᖰᒡᒧᐧ ᕼᒼᖽ (Blackfoot / Niitsítapi ᖰᒡᒧᐧᒡ), Secwepemcúl'ecw (Secwépemc), Tsuut'ina**,** Michif Piyii (Métis), and Mountain Métis people. Finally, we are grateful to Martin Sharp for illuminating discussions.





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





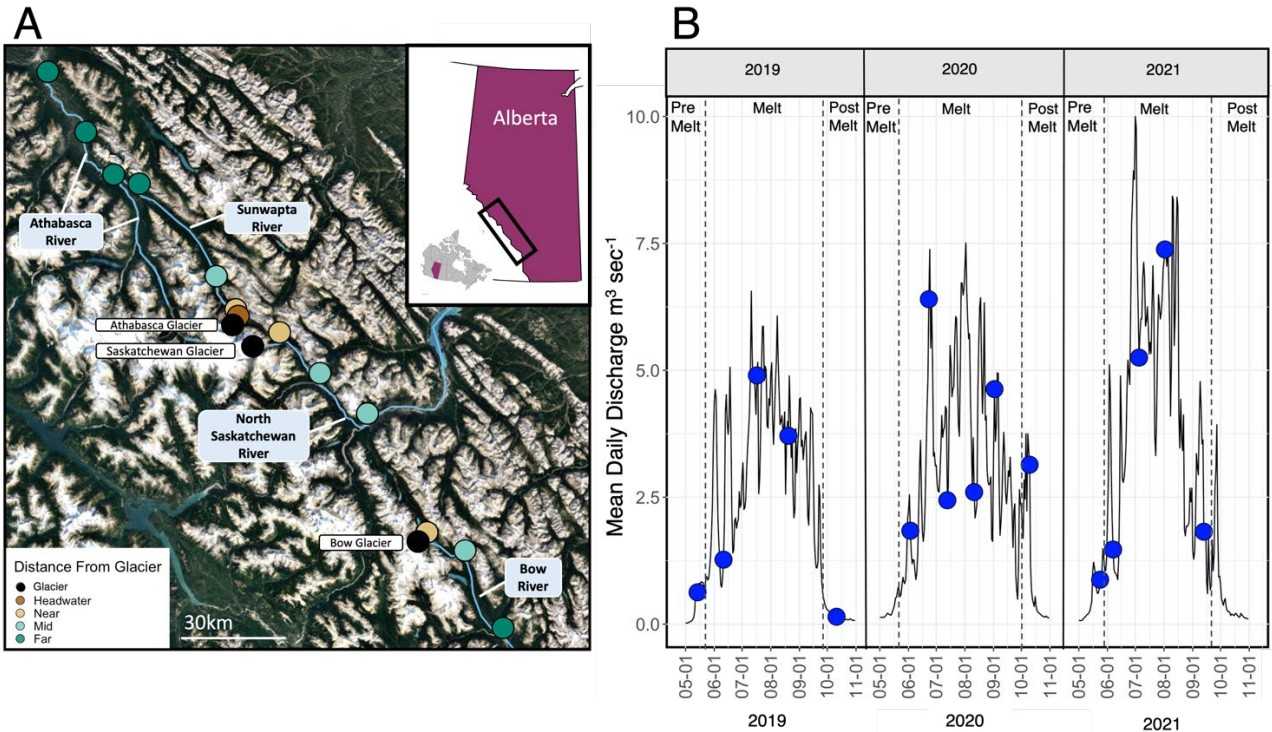

**Figure 1:** (a) Sampling locations of study sites on the Bow North Saskatchewan, Sunwapta, and Athabasca Rivers, in Alberta, Canada (Google Earth © 2022). Locations are coloured by distance range, binned as headwater (0.3 km downstream), near (3-7 km downstream), mid (18-35 km downstream) and far (40-100 km downstream). The inset map shows the location of the sampling region (black box) within Alberta. (b) Hydrographs of open-water discharge measured at the gauging station three km downstream of Athabasca glacier (station 07AA007, maintained by Environment and Climate Change Canada) from May to November 2019–2021. The location of the hydrologic station corresponds to the Sunwapta "near" site. Sample collection dates are demarcated by blue dots.





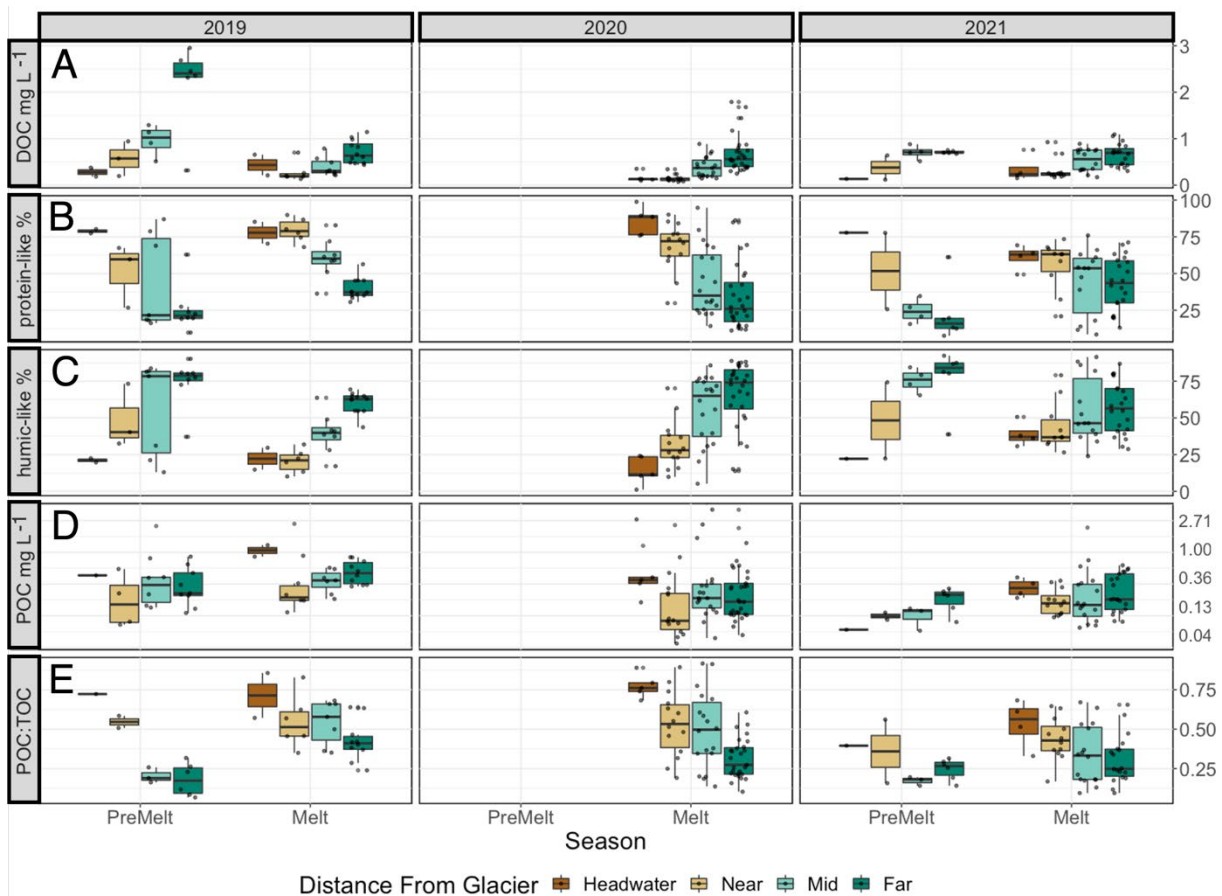

**Figure 2:** Boxplots of: (a) DOC concentration, in mg L$^{-1}$ (b) relative percentage (%) of protein-like DOM (c) relative percentage (%) of humic-like DOM (d); POC concentration in mg L$^{-1}$ (shown on a log scale); and (E) the ratio of POC to TOC (POC + DOC). Data are shown for pre-melt and melt seasons during 2019–2021 across distance ranges. Pre-melt samples in 2020 are missing due to a delay in the field season start because of the COVID-19 pandemic. The boxes represent the inter-quartile range, the black line represents the median value, and individual dots show all collected samples.





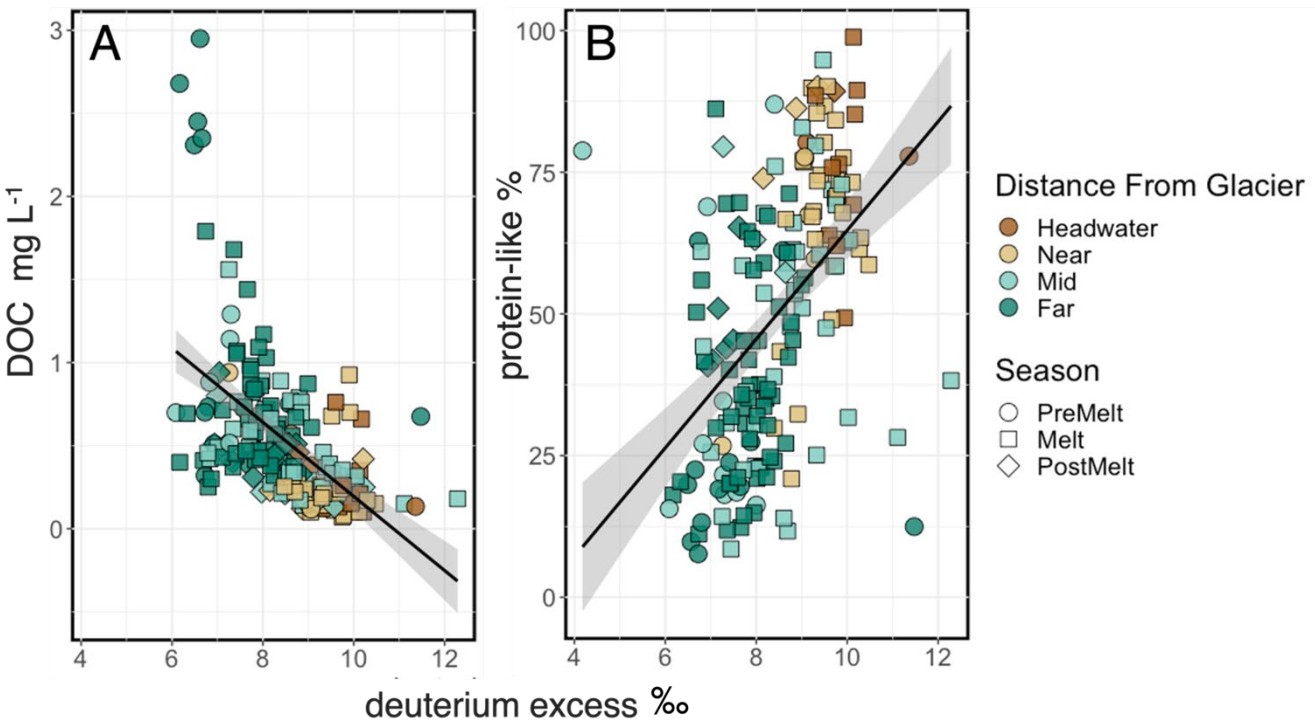

**Figure 3:** The relationship between (a) DOC concentration (mg L$^{-1}$), and (b) the relative percentage of protein-like DOM (%) and the calculated percentage of deuterium excess within streams. Colours represent distance ranges and shapes represent hydrological period. The black line shows a linear fit (DOC = -0.2*dexcess + 2.33, R$^2_{adj}$ = 0.28, p < 0.0001; protein-like% = 10*dexcess – 31.2, R$^2_{adj}$ = 0.23, p = <0.0001), with the grey shading demarking the 95% confidence interval.





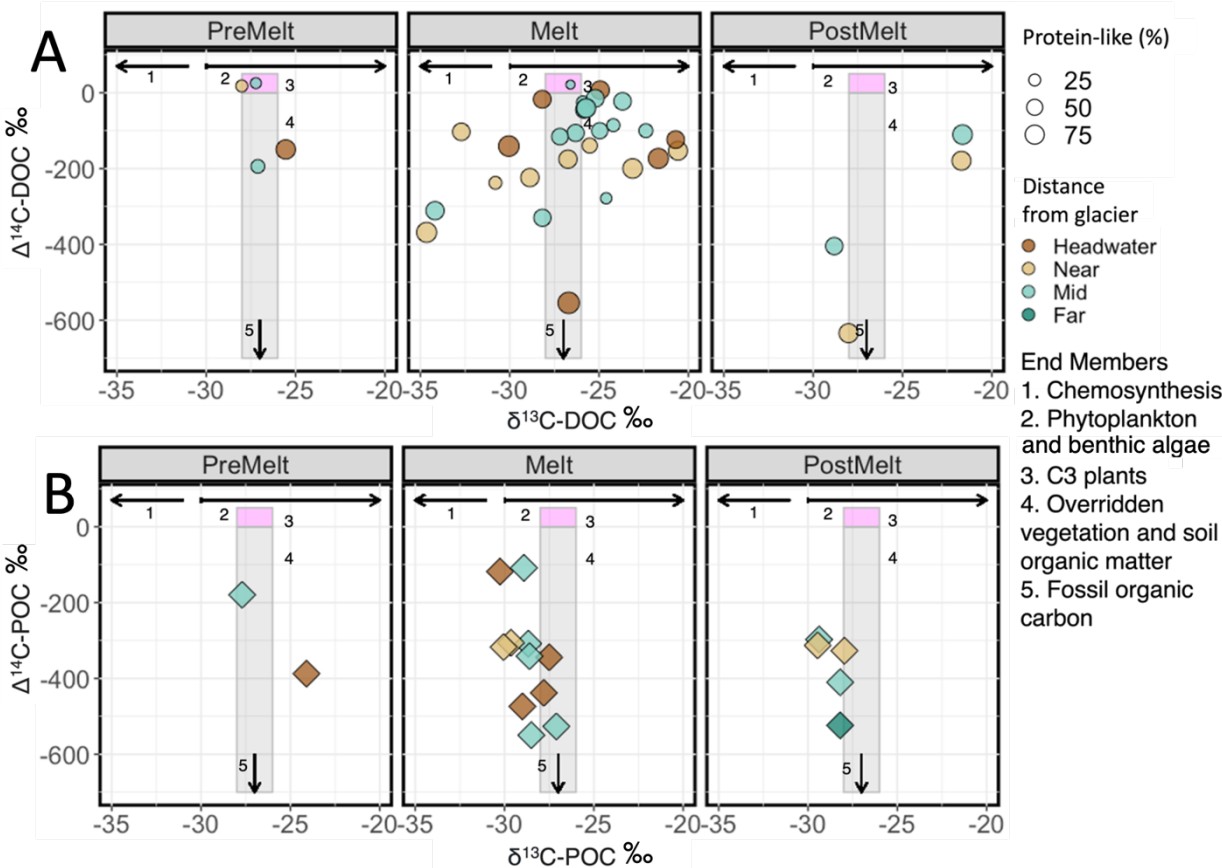

**Figure 4:** $\Delta^{14}C$ versus $\delta^{13}C$ values for (a) dissolved OC with size indicating the percentage protein-like DOM (%) in each sample and (b) particulate OC. Note that symbol size variation is not applicable in panel B. Different colours represent distance ranges. Numbered (1–5) pink and grey boxes and black lines represent literature isotopic ranges for various endmembers (Table S13).



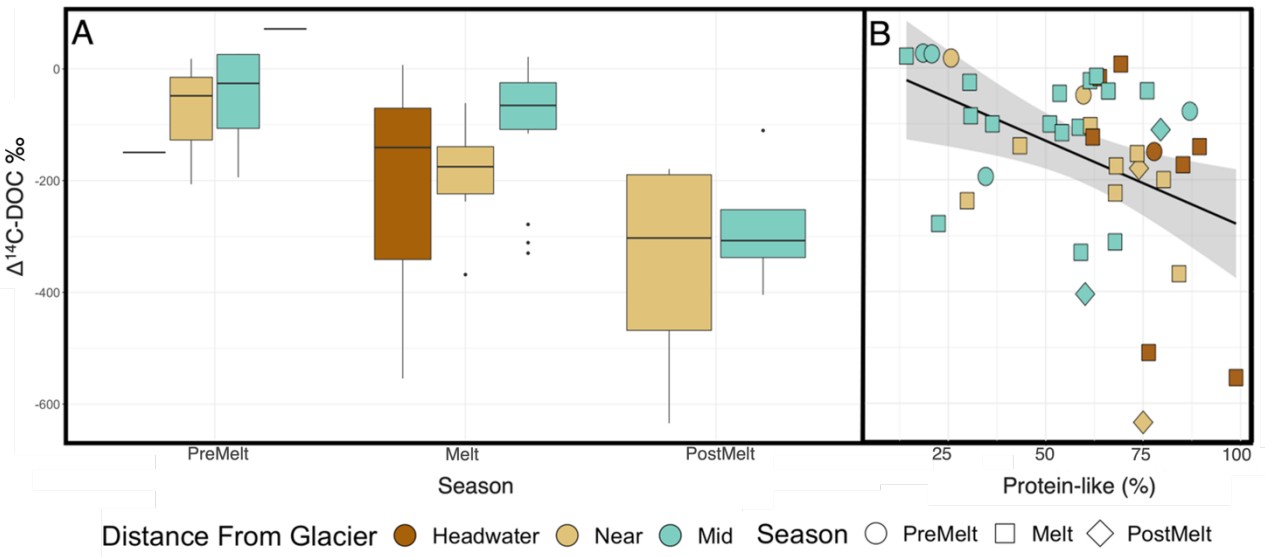


**Figure 5:** (a) Boxplots of stream $\Delta^{14}$C-DOC values for pre-melt, melt and post-melt hydrological periods grouped by distance range. The boxes represent the interquartile range, the black line represents the median value, and the points represent outliers. (b) The correlation between stream $\Delta^{14}$C-DOC and the relative percentage of protein-like DOM (%). Colours represent distance range and shapes represent hydrological periods. The black line shows a linear fit (% protein-like = -303.72 $\Delta^{14}$C-DOC + 21.70, $R^2_{adj}$ = 0.15, p = 0.006), with the grey shading demarking the 95% confidence interval.







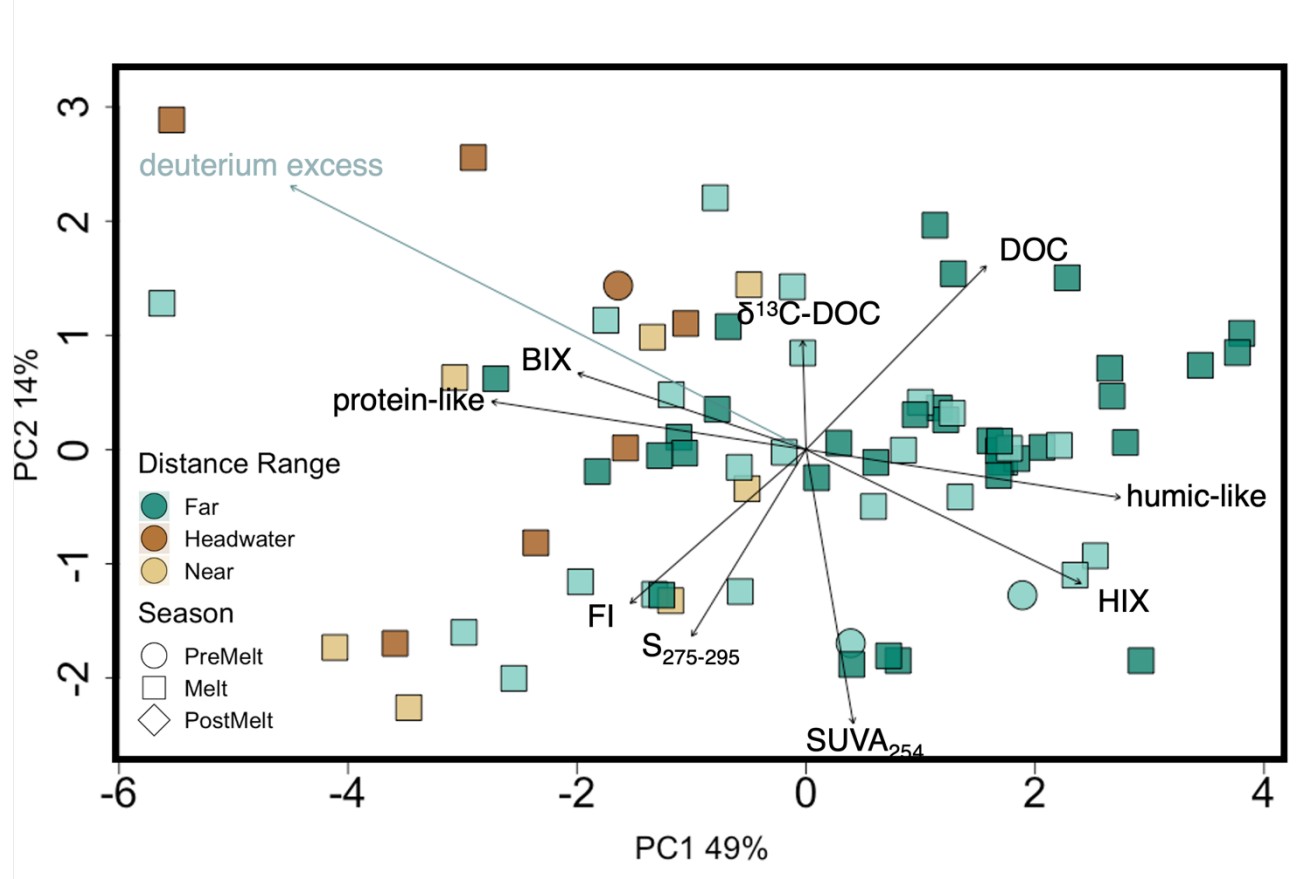

**Figure 6:** Principal component analysis of dissolved OM parameters (DOC concentration, $\delta^{13}$C-DOC, $S_{275-295}$, $SUVA_{254}$, BIX, FI, HIX, protein-like DOM, and humic-like DOM). Deuterium excess (blue text) is a passive overlay. Colours represent distance range, and shapes represent hydrological period.



**Figure 7:** (a) Venn diagram showing the number of identified ASVs at the headwater (n = 4 samples), near (n = 20) and far sites (n = 42). (b) Bar plot showing the relative abundance of ASVs that are unique to each distance range, those ASVs identified in all three (headwater, near, far) distance bins ("core") or those that are shared between two distance bins ("other") in each sample. Sample abbreviations on the x-axis indicates river (Sunwapta (S), North Saskatchewan (N), Athabasca (A)), distance downstream (km), and sample date (as mmyy).





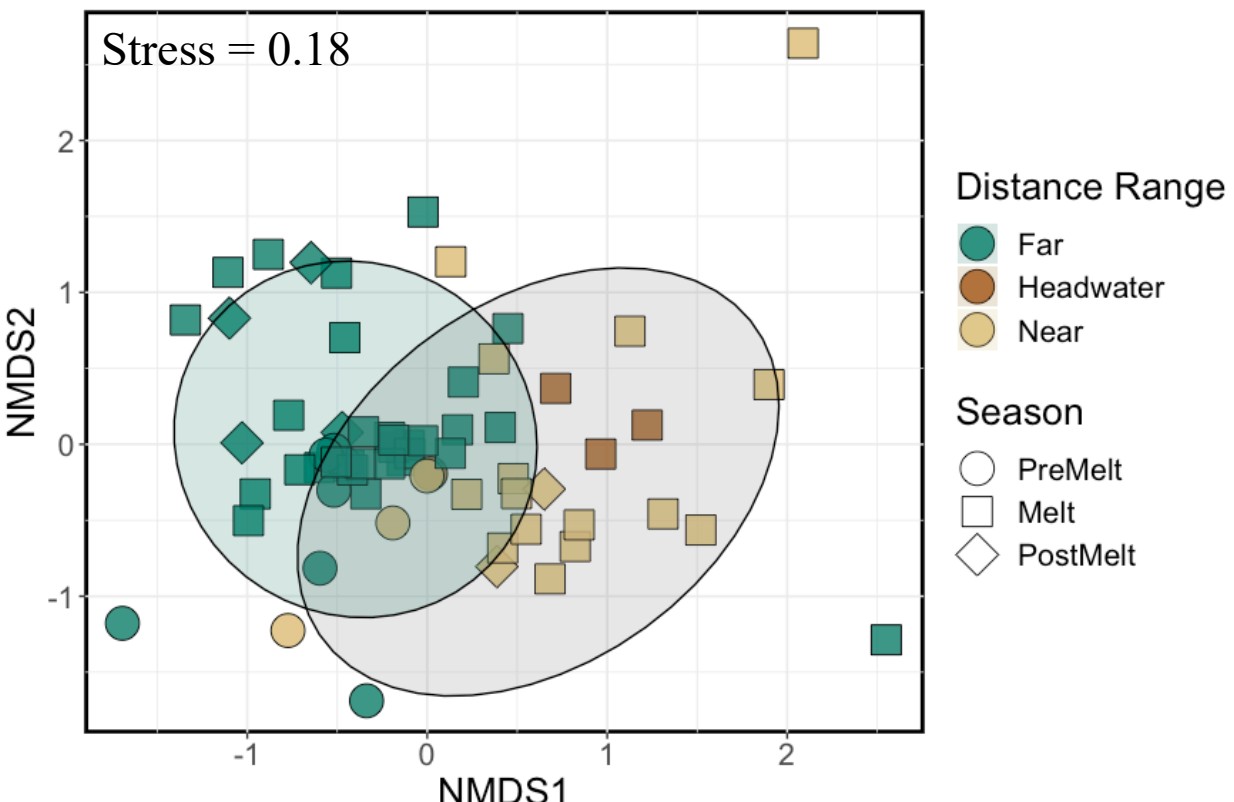









**Figure 8:** NMDS of microbial community composition of stream samples. Shapes represent different hydrological periods while colour indicates distance range. Shaded circles represent the 95% confidence interval for significantly different groupings (pairwise perMANOVA p < 0.01 (holms adjusted), R2 = 0.06-0.09, see Table S12 for comparisons).

**Figure 9:** Redundancy analysis of microbial community composition at the headwater, near, and far sites constrained by environmental variables, with black arrows showing significant explanatory variables as determined by a backward selection analysis (see methods). The percent variance explained by the first two RDA axes is shown, as is the unadjusted R2 and adjusted R2 values of the variance explained in microbial community composition by our included environmental variables (see methods). Colours represent distance range and shapes represent hydrological period. Black crosses in the centre represent individual ASV scores.


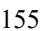

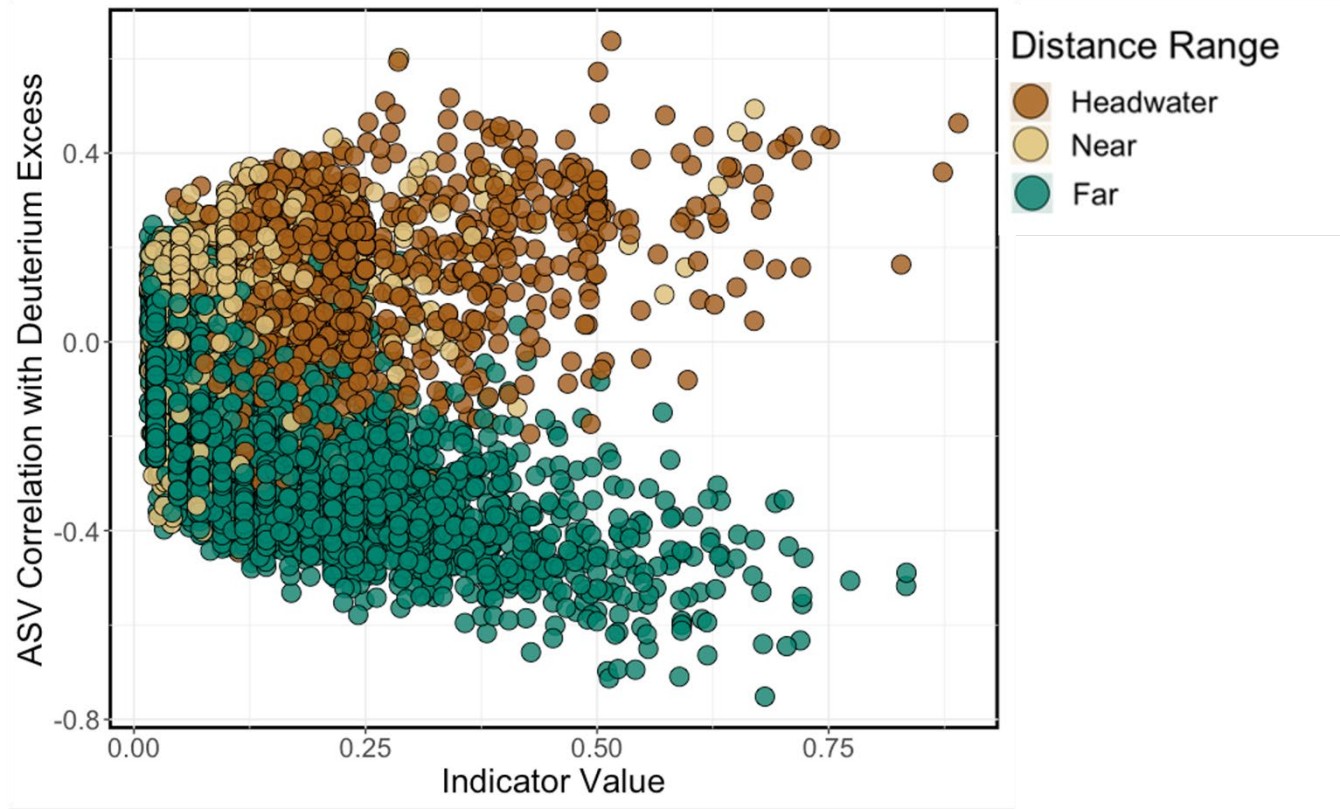

**Figure 10:** A scatterplot of the relationship between the Spearman's rank correlation coefficient for ASV relative abundance and deuterium excess, and ASV indicator species value. Brown, beige, and green circles represent the ASVs resolved in the headwater, near and far samples, respectively.








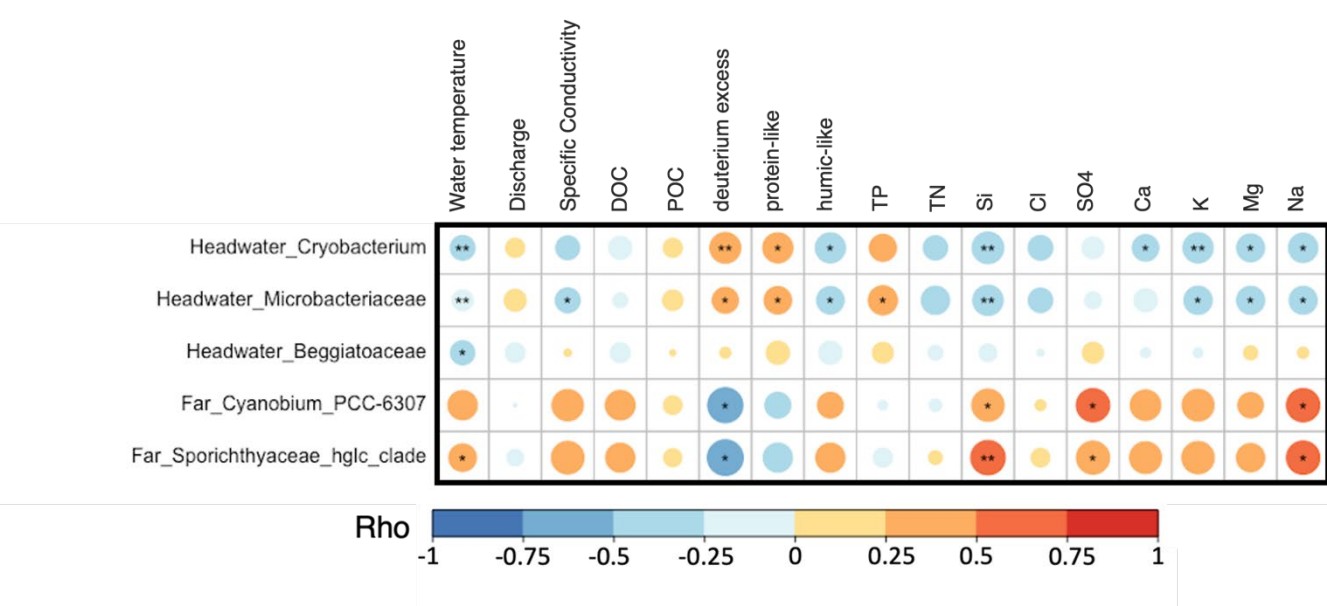

**Figure 11:** Spearman's rank correlations (Rho) between the abundance of strong microbial indicator species (IV > 0.80) and environmental parameters. Stars represent significance level, with p < 0.05 (*), and p < 0.01 (**) indicated.