# Peer review of "Shifts in organic matter character and microbial assemblages from glacial headwaters to downstream reaches in the Canadian Rocky Mountains"

_Biogeosciences, 2023_

## Author Comment (AC1)

**Reviewer 1:**

In this manuscript, the authors have addressed an important question, effectively how glacier fed streams change longitudinally, which I fully believe is in the interest of the Biogeosciences readership. The data themselves are very interesting (with some nice longitudinal patterns), as the analytes chosen are good proxies for glacier processes and for relevant biogeochemical cycles. The paper is overall also quite well written, and the figures well illustrated. However, in carefully reading through the methods and results sections, I have three major comments: 1) The statistical analysis can be improved. The issue to me is that there are three streams, sampled at different points in the hydrograph (premelt, melt, postmelt), at 3-4 different transects, over three years. This is already a lot of categories for the dataset size which makes interpretation tough. Also problematic is that the categories are also not neatly defined: the 'longitudinal' categories encompass quite some different distances, and there is uneven replication and temporal representation (mostly in melt versus pre- or post- melt, the distinction between each seems arbitrary). Another issue is it seems that the data for the three streams are lumped together in the analyses. Can this really be justified…..wouldn't it be better to analyze each glacier stream individually? As a result, I think that there could be quite a problem with spatial and temporal autocorrelation, and I think data independence is an assumption with the utilized analyses. I have read through this paper several times now, and I must admit, I don't have an obviously better solution given the sampling design, but one possibility could perhaps be some mixed models (ex GLM or GAMM etc) that account for the different categories, as well as the possibility of making some of the categorical variables continuous (i.e. distance from the source glaciers) to improve subsequent interpretations. 2) Another general issue is the justification for why these things are done in the first place….what do the authors hypothesize, particularly with regard to each of these tested categories? The stated rationale of seeing how they change with distance downstream can be much elaborated upon. 3) Finally, I was a bit disappointed in the lack of information regarding the microbiological analyses and results (in more detail below), which are really necessary to assess the quality of the data and interpret the results. Therefore, I challenge the authors address these three major points in their revision. Some more detailed points follow by line number.

We would like to thank the reviewer for the thoughtful and thought-provoking review of our manuscript. In relation to the specific points raised above:

(1) Thanks for this statistical suggestion. We had considered a mixed effects model at the outset of manuscript preparation, but chose to move forward with the "binning" approach, mostly for data visualization reasons. We can move to a mixed model for our analyses. We have tested a model that includes distance and melt season as fixed effects, and year and river as random effects, and can confirm it is workable, and does not change the major conclusions of the manuscript.

(2) We will add specific hypotheses or objectives at the end of the introduction, as also suggested in the detailed comments below.

(3) We will add more details on the microbial analyses, as further laid out in the specific comments below.

Line 13: be careful here…..not sure that this OM is necessarily structuring water column communities if they were just exported there, and especially given the result that mass effects seems more prevalent than environmental filtering

We will rephrase the wording on this point.

line 18: microbial communities inhabiting what? There are a lot of habitats in streams, and I think this detail would be of interest to those reading the abstract. As I have already read the paper, I feel like these are probably not communities as all, but microbial assemblages in transport. I think this should be clearly stated here, and I make further remarks on this below

Samples for microbial community analyses were collected as described in the previous sentence. We do not expand in this sentence to allow for brevity in the abstract.

line 21: although I never really read any names of the putatively chemolithoautrophs or cold adapted taxa….mostly just phyla and other high-order taxonomic names
This point in the abstract specifically refers to the indicator species. We provide characteristics of the indicator species in the Discussion (L515), but see how this could be overlooked given that it's in the middle of a paragraph. We will add this information to the indicators species section of the results (Sn 3.2.1) so that it stands out, and then refer back to this information in a more general way at the current L515 of the Discussion. Given the need for brevity in the abstract, we will refrain from adding more specificity here.

line 22: while I don't dispute the role of glaciers in 'seeding' headwaters, im not sure the data really show this…..you don't have glacier endmembers, and you cannot prove that the microbes were actually alive or living there, so there is not really a 'smoking gun'….just some precision may be needed in the language
Our furthest upstream sampling point was ~300 m from the glacier termini, so the vast majority of water (certainly during the melt season) was glacially-derived. We certainly agree, however, that mass effects appeared to dominate; we will adjust our wording slightly here to ensure precision.

line 23: probably all could be argued to be indicators of water source
True! This qualifier was added at this point in the text because deuterium excess (i.e., the isotopic composition of water) is not really an "environmental condition", per se. We will rephrase as: "…environmental conditions (including water temperature; POC concentration; protein-like DOM; and deuterium excess, which we use as an additional indicator of water source) …"

line 29: 'complex ecosystem responses' sounds pretty vague….could you make any more specific predictions?
Agreed. We can modify the text at this point to refer to loss of glacial seeding, and concomitant "terrestrialization" of organic matter chemistry that may also cause change in microbial community composition.

Line 43: 'supraglacial' and 'subglacial' would be related to glacier tops and bottoms, respectively. Marginal channels are not found on top or beneath the glacier (unless I am misunderstanding something).
We will rephrase this sentence as follows:
*As glaciers melt, the formation of supra- and subglacial channels can cause meltwater to be routed on top of, through, and/or underneath, glaciers.*

Line 49: This process also helps to explain a lot of the differences in your analyzed variables, which should probably be discussed to justify their collection
Agreed. We will alter this sentence to refer to both hydrology and biogeochemistry.

line 56: the POC is from crushed rock? Couldn't it also be from the overidden vegetation?
We will switch our wording here to read "can be predominantly derived from", and add a reference to Cai et al. (2016) that quantifies this effect. At these Alaskan sites, glacial-origin POC was found to be predominantly petrogenic in origin, although certainly overridden vegetation would also contribute to this pool.

Line 64: agreed that these are important variables for benthic communities, but do you think the same is true for the organisms in the water column?

Yes, we do think so - we would argue that light penetration (particularly in sediment-laden streams) and flow (associated with sediment, but also generation time) are important for water column communities, as well as those that are associated with the benthos.

Line 73: this seems to be a specific number……from my experience, glacier surfaces are remarkably heterogeneous, which may be good to note.
Yes, fair comment – we will broaden the scope of this statement, which is specific to one publication.

line 78: although there is only microbial data for the headwaters and far reaches, and none from Bow
It's true that we only present data from headwater (300 m downstream of glacier), near (3-7 km downstream), and far (50-100 km downstream) sites, which is a decision that we made for data visualization purposes - it's such a rich dataset!  Reasons for excluding the Bow microbial data are described elsewhere. Re-reading this sentence, we feel it is still "true" to the data analysis  that we did, and the dataset we present.  We will swap "rivers" for "reaches" to ensure there is no confusion on the scope of the work.

line 81: how specifically can this work inform us about how glacier loss will impact microbial diversity?
At this point in the manuscript, via either (1) loss of source communities, or (2) changes to "environmental" condition (water chemistry, temperature, etc.) that drive changes in microbial composition. Hopefully with surrounding edits, this will be clarified.

line 84: this is repeated from the first paragraph, line 34 I think….
Thanks for catching this; we will edit to avoid redundancy.

Line 90 to line 93: While this doesn't bother me, it seems a bit strange to give the results of the paper in this last paragraph…..this could potentially be skipped. What would be nice to include, however, would be some specific hypotheses that could be tested. What exactly did you predict would happen to the OM and why? How should microbial communities change with space/time and why?
We are happy to provide hypotheses at this point in the paper, though may still retain some key take home points to make sure they're clearly conveyed to the reader.
At the outset, our hypothesis was that we would see clear variation in chemistry with movement from glaciers to downstream reaches, and that this would be structure microbial assemblages, thus causing these to also vary along this gradient.  Setting this hypothesis up clearly at the outset will help us to reinforce some of the key take-aways of the paper (i.e., evidence for mass effects) and thus strengthen the paper overall - thanks for the suggestion.

General comment for introduction: The introduction is written nicely, but many of the papers used as citations are from work on the Greenland Ice Sheet, which is really an enormous piece of ice with its own very special characteristics. I know nothing about the glaciers which were the basis for this study, but I encourage the authors to evaluate whether or not these citation are applicable in the context of their study system, and if some papers dealing with smaller glacier might be more relevant to cite in some cases.
Agreed that the Greenland Ice Sheet is very different from the mountain glacier system that we are studying. In the Introduction, we draw from studies conducted in Greenland, but also from glaciers in the Alps and Alaska. There has actually been very little work on carbon biogeochemistry specific to Rocky Mountain glaciers, which was part of the impetus for our study!  We prefer to keep this diversity of references, because they show the consistency of what one might expect regarding glacially-sourced organic matter composition, for example, and thus helped to set up the initial hypotheses for the study.

Line 96: I see that there is a hydrograph for one of the streams, but how do the discharges of the other two streams compare with this one? Could be much different given that different size of the icefields mentioned here, yet there is no way to tell. This is an important consideration given that all three rivers were lumped together in the analyses, though my feeling is that the effect of 'stream identity' should be taken into consideration.

Please see below (response to comment at L118) about availability of hydrographs during our study year, and consistency of discharge across sites.

Following the comments of both reviewers, we have changed our analytical approach to more specifically account for river identity (see more detailed comments above on our mixed effects model).

Line 103: be careful with the word "evolve", since it has quite a specific meaning in the biological sciences

We will switch "evolve" to "varied" at this point in the text.

Line 105: These glacier distance binnings seem arbitrary to me. Is there some ecological reason that we could expect changes with distances of these magnitudes, or are they indeed arbitrary? A concern for me is that the difference in the far sites (100-40=60km) is actually greater than the distance of the first three other categories (0 to 35 km), as well as 1/3 of the 'far' category itself. I guess my question is if these categories make ecological sense given this, or would it make more sense to use distances from the glacier as a continuous variable, in which case the 'realistic nature' of these categories would no longer matter?

These distance bins were selected *a priori* to reflect the substantial difference in watershed composition between bins, as follows:

- The headwater site was immediately (a few hundred meters) downstream of the glacier termini, and thus received almost all melt-season runoff from glacial sources. Within its watershed, this site had had 0% forest cover, 50.7% coverage by snow and ice, and 48.8% coverage by rock and rubble
- Near sites were 2 - 7 km downsteam of glacier termini, and above the point of substantial forest establishment in the catchment, but with a large enough catchment area that source waters would be from variable sources. These sites all had less than 5% forest cover.
- Far sites (greater than 50 km downstream) were at sites with significant forest (and thus soil) development adjacent to the sampling site, and with 25% forest cover within the broader watershed (i.e., including upstream, mountainous, reaches).

We will add these details to the current Table S1, and flesh out this *a priori* rationale at this point in the text.

However, given the similar comments from both reviewers, we will switch to a mixed effects model for our chemistry analyses (i.e., move away from the ANOVA approach), which will include distance as a continuous variable. We agree with the reviewers that this is a more suitable statistical approach, though - after testing it out somewhat - we can confirm that it does not change the overall conclusions of the paper. This will mean changing most box-plots figures to scatter plots with distance plotted as the independent variable.

For the microbial analysis, we will continue to use these distance bins (e.g., as an input to perMANOVA in the NMDS, the indicator species analysis, and the Venn diagram and associated analyses); we have experimented with other approaches and data visualizations, and keep coming back to this being the best way to analyze and show differences among sites.

Line 108: I think "stream" works, but its then a bit weird then that you used 'river' in the title

For clarity, we will rephrase the title as follows:

*Shifts in organic matter character and microbial community structure from glacial headwaters to downstream reaches in the Canadian Rocky Mountains*

Line 113: how many of these samples were taken in December and January? There are not a whole lot of winter glacier fed stream data, so these could be of extra interest…. Where they different than the others?

Unfortunately they are buried within the rest of the other data and its not possible to see them. They also don't seem plotted on the hydrographs. Thus, despite reporting them here, they seem invisible in the paper.

Yes, thanks for pointing out that these samples seem to disappear in the manuscript. We sampled in December of 2019 and January of 2021 (i.e., during under ice periods after the first and second open water sampling years). Of our sites, we were only able to access three in 2019 and four in 2021, as a result of poor site accessibility and overall safety considerations. We can modify figure 1 to add these data points beside the hydrograph: they are not shown in the current figure because they fall outside of the recorded hydrograph. Many of the results from these sampling periods were "below detect", but we will add some additional description about these samples to the manuscript.

Line 115: though the sample coverage over these three periods is very uneven, and most samples are from the melt period. Should put some sample numbers here to give the reader an idea of how many samples were collected at each period. Also, do you have a feel for how well the Athabasca Glacier hydrograph corresponds to the hydrographs of the other two streams?

We will add the number of sampling campaigns in each period at this point in the manuscript.

The second question is a challenging one, because the Sunwapta outflow is the only one that is gauged. However, the three glaciers are in quite close proximity to each other and so experience relatively similar climate (i.e., presumed start to melt season). And, we know from our own field observation that the onset of more rapid flow (i.e., associated with the summer melt season) was similar across our sites. The outflow of the Bow Glacier was gauged in 1973 and 1975 (since discontinued); of these two years, 1975 had a much longer (but still not full) melt-season record. We provide a figure that superimposes the Bow and Athabasca glacier outflows below, illustrating synchronous pulses in streamflow that presumably reflect relatively homogenous temperature and precipitation patterns within this region.

[Figure]

*Figure 1:* *Measured discharge of the Bow Glacier Outflow (Station number 05BA009), and Sunwapta River at Athabasca River (station number 07AA007) during the open water season of 1975.*

Line 118: instead of arbitrarily using 1 m3, couldn't you use your hydrological/chemical data you collected to determine if the subglacial channels have opened or not? By using arbitrary cutoffs as you have here, there is a good chance that you will get results from your analyses without the stated ecological relevance. Also, what kinds of differences might you expect to appear in your data as a function of season? These hypotheses are not clearly identified….

The 1 $m^3$ threshold was chosen following close inspection of the Sunwapta River at Athabasca Glacier hydrograph (i.e., the discharge monitoring station most immediately downstream of this glacier). Although the 1 $m^3$ may appear to be arbitrary, the cutoff was chosen because it represented the inflection point characteristic of meltwater contributions to flow. We will rephrase in the manuscript to clarify this point; thanks for pointing out that we had neglected to do so.

Line 125: thus sample year should probably also be accounted for in an eventual model
See above for changes to the statistical approach, and our planned switch to a mixed effects model for this component of our statistical analysis. THis approach will include sample year (though we note that our original approach via ANOVA did also include year as a factor).

Line 126: would be nice to know how many samples were collected from each of these categories. I know that some of this information is in Table S1 (i.e. sites as a function of distance), but it would be really good to have some of these numbers here too. From looking at the plots, its seems that the pre-melt and post-melt periods are vastly undersampled compared to melt periods, yet there is no way to really know this except by looking at figures.
We will add this information at L115 (see above).  It is true that we have many more samples during the melt season;  however, this is by virtue of the fact that we sampled every 3-4 weeks from the onset of streamflow through until September, and then twice (October and under ice) thereafter. Thus, it's really just that the dataset is incredibly rich, particularly during the summer months.

Line 133: should probably also put the charges on the trace metals since there are charges on the other ions (also on line 200). Also, why exactly were these measured? There is no hypothesis stated, and the data arent really discussed anywhere else? Same could be said for the major ions for that matter.
We will add charges. We include these data to enable a full assessment of possible environmental controls on microbial community composition (see also our response below).  A more thorough work-up of the trace metal and nutrient data is provided in a recent publication by Serbu et al. (2024), which we can now reference directly in our manuscript.

Line 134: I think its common to acid wash bottles for nutrients too, although I realize this wasnt the focus on the work, and the difference in results likely not large. What does the citranox do?
The analyses were run in a CALA (Canadian Association for Laboratory Accreditation)-certified lab, which is - among other analyses -  certified for the analysis of nutrients  in freshwaters. The lab has tested collecting nutrients using new, unwashed bottles of this specific type, and found that this gave better (lower variability, lower and more consistent blanks) results when compared to reusing (and washing) bottles. Our procedures followed this SOP.
Citranox is a detergent which is phosphate free, and is commonly used to keep labware and equipment clean in many environmental laboratories. It is an extra cleaning step prior to acid washing,  which does not introduce phosphate contamination, and is able to remove any metal oxides, salts and inorganic residues that may remain in the sampling bottles, without leaving behind any contaminating residues.

Line 150: how much water did you filter for the microbes, and what exactly does 'prepared' mean in terms of the bottle?
We collected 2L for each microbial sample, and filtered "until refusal" of the sterivex filter. In some cases, filters were loaded before the full 2L was filtered. We will add this detail to the text.
Prepared just means cleaned, as described in the previous paragraph. We will switch our wording for clarity.

Line 200: calling dSi a nutrient is contentious. Also, what were the limits of detection for your nutrient analyses?
We prefer to include Si in the list of nutrients because it can of course be limiting in some situations. We will add LOD details to the supplement.

Line 204: could give the nutrient methods numbers here as well

We can add more detail on the nutrient methods; as described above, all analyses were conducted at lab that was CALA certified for these analyses.

Line 215: do you have a citation for the modifications of the protocol? Or do you have some data on what kind of improvement you can expect? Just that this might be of interest to others doing this kind of extractions with these kits

We do not have a citation for this modification: this was a modification based on the expertise of Dr. Josh Neufeld, who had advised one of our authors when this protocol was first implemented (2016: much earlier than this study) for similar work that dealt with highly oligotrophic waters. The results from this modification suited our needs (DNA yields were often >1 ng), and we carried forward this technique for this study.  For context, there was concern that 5 minutes at such a high temperature may degrade any microbial DNA that had been captured outside cells on the filter, and/or would not be as efficacious in lysing cells uniformly.

line 233 and elsewhere: I think that there should be quite a lot more information on the microbial data. For example, how much water was possible to pass through the sterivex in the end? How much DNA were you able to get from the filters following extraction? What kind of depth did you sequence to/how many reads per sample? Were they rarefied for the analyses? I think these are pretty important things to report given that these are often difficult habitat types to work with…..

Thanks for this: we will add more information on the microbial data, either at this point in the methods, or early in the Results, and agree that more detail would be helpful. To answer specific questions here:
-Details on filtration are is in our response above (up to 2L filtered)
-Measured DNA from filters ranged from below detection to 20.8 ng/ul; when DNA concentrations were very low (determined by observing <10ng/uL on Qubit following DNA extraction), we added 5uL of template DNA (default is 2.5uL), along with BSA to coat the sides of the PCR reaction tube to maximize DNA available for PCR, adjusting for total water added to the reaction.
-We had as many as 300,000 reads per sample, but also with large variability; samples with less than 5,000 reads were excluded because they had not reached a plateau, indicating that sequencing might not have captured a good representation of species (see Figure 2, below).

-We chose not to rarefy because of the well-known debate on this topic. However, we did run all microbial analyses with a rarefied dataset, and this did not change our overall results.

[Figure]

**Figure 2:** *Rarefaction curves to illustrate the number of amplicon sequence variants detected as function of sequence depth on individual samples.*

line 235: what do you mean by "low abundance ASVs"? Like….you removed rare ASVs? At what threshold….why??
Here, we were actually referring to ASVs that are present as singletons. We will reword for clarity.

line 242: not a big deal, but seems that the DOM ab/fluor should go with the DOC methods above rather than the microbial data
Looking at this again, we think it makes more sense to move this text to section 2.4 (Data analysis), since it focuses on data post-processing.  We will reconfigure in the revised manuscript.

line 255: why did you choose three way ANOVA rather than keeping these as continuous variables? If you made mixed models or similar, you could control for the effects of stream, season, etc, while also using the data directly as continuous variables rather than making arbitrary categories for downstream distance, discharge, etc
See comment above – we will move to a mixed-effects model, and have tested to confirm that this is workable for our dataset.

line 260: though I think deuterium can also differ from glacier to glacier…..would be possible that these numbers different between the headwaters of the three streams. Also, why plot this as a passive overlay on the ordination? Should explain this….
Yes, it's possible that d-excess varied between the two glaciers, though this is a well used metric for water source in glacial and other high elevation studies.

We plotted as a passive overlay because the ordination focused on organic matter composition, and so was confined to these specific parameters. The passive overlay of d-excess is to allow the reader to make some conclusions about water source (acknowledging that this is not perfect …), and how DOM composition varies with this parameter.

Line 265: I think it should still be possible to compare and contrast headwaters and downstream samples with the middle sites included…..not sure I understand this justification….I would probably just include all of the data, no?
The reviewer is correct that including the mid sites does not change the conclusions of the manuscript in any way.  However, it does improve our data visualization quite a bit. After iterating on our data presentation a fair amount, we decided to move forward with this presentation, and would prefer to stick with this approach. We do note that all data are included in our dataset upload to  NCB, enabling others to explore these data further.

Line 266: maybe I am old fashioned, but I like to know the actual number of ASVs rather than using shannon for alpha diversity
We specifically chose to calculate and present diversity (i.e., via Shannon Weiner) rather total ASV count because: (1) our focus was on diversity, rather than richness, and (2) total ASV count is of course affected by the amplification process, and so not a "true" estimate of taxa richness, in the way that estimates of richness for other taxonomic groups are understood to be.

line 268: is it necessary to square root transform AND calculate bray curtis distances?
Yes, it is preferable (and rather a common practice, i.e: Zorz et al., 2019, Laporte et al., 2021…) to do both on a microbial dataset, usually because microbial community data does not represent a normal distribution amongst all samples (large abundance of microbial members in a few samples, but several samples with much smaller abundance), which can be an effect of sequencing depth. This can skew beta diversity analyses, such as what is visualized by the NMDS, if we do not first standardize the data. Transforming the data by Hellinger transformation allows us to standardize the microbial community dataset by assigning lesser weight to samples containing low counts and/or 0 of particular microbial amplicon sequence variant (ASV), which is a common attribute of microbial community data. The application of a bray curtis distance then determines how different the abundances of microbial members are between samples more accurately, versus if we were to simply use a bray curtis dissimilarity matrix without standardization.

Line 269: so you identified clusters on the figure and then tested the clusters with permanova? That sounds a bit self-fulfilling to me…..wouldnt it be better to test hypotheses using the data? Also, there are likely far too many parameters that were included in the dbRDA, and its not clear why most of them were included. Perhaps you can provide some justification for why some of these are here? For example the trace metals? How many factors were left after the highly correlated ones were omitted?
As described above, we identified clusters *a priori* using distance-based bins (Figures 7-9), and also (Figure S6) by river and year. PERMANOVA was then applied to these *a priori*  clusters. We will certainly ensure that the reasoning for the distance bins is better articulated in the revised manuscript, following on comments from both of the reviewers.
Similarly, we did not want to exclude any environmental parameters a priori in the dbRDA, because of the broad suite of factors that is known to affect microbial communities. In this case, we did not feel that we had good *a priori* reasons to exclude environmental parameters, and so included all data that were available.  The poor explanatory power of the RDA perhaps reinforces this decision - if we had run it with just a subset of the environmental data that we had available, we likely would have chosen to 'circle back' to see if we were missing something. We had 12 parameters remaining after highly correlated parameters

were removed, and additionally used a backward selection approach to assess significant contributors to model fit, rather than including all parameters in the final model.

Line 278: Should probably correct for multiple testing, no? Also, what exactly will testing co-variation between indicator species and environmental parameters tell you? My feeling is that indicators of upstream will just be correlated with things more likely to be characteristic of upstream sites, like low temperature for example
Following comments from both reviewers, we've decided to exclude this figure from the manuscript.

line 280: well…..its the R statistical environment, which uses the R programming language
We can change "*with the R programming language*" to "*within the R statistical environment*".

Line 287: what kind of test does this p value correspond with? Also, there is likely to be big differences in sample sizes between categories…..were there differences between rivers, and if so, can you justify lumping the data together?
This p-value corresponded to the ANOVA, which included river as a factor. However, this presentation will switch with the move to a mixed effects model.

Line 294: although there wasnt so much pre-melt data for the other years…..
Comparatively, perhaps, but overall we would argue that this is quite a rich data set!

line 297: 'largely non-significant trend'…..probably it was just not significant
This wording will change with the move to a mixed effects model.

line 319: could be due to the high/low points on the hydrograph during the meltseason, which could be related to greater sub/supraglacial contributions
Thanks for this insightful comment - the more enriched headwater PO14C sample was taken on 23 June 2020 (i.e., second sampling point in 2020), when flows were quite high.  The more depleted samples (across pre-melt and melt) were taken across a variety of flow conditions. We don't see a clear relationship, but this is a very good question.

Line 338: These phyla are literally everywhere, thus making this sentence not very informative. Not that it is necessarily bad to mention these, but I would focus on lower taxonomic levels….
Agreed. We will keep these two introductory sentences that focus on phyla, but bolster the text that follows as described below.

Line 343: How was the core defined? There are many many ways to do this, and it would be good to know what were your assumptions….probably in the methods. Also, there are two interesting things that come up with this: First, you call it a community, but im not sure this is an accurate term to apply to these microbes, given that almost all of them are probably being passively transported, and therefore not 'residents'. Also, why do you think there should be a core…..what are your hypotheses here….how is this core maintained? In any case, a core of 1,409 ASVs tells us very little because we have no idea who they were, how many reads were generated per sample, and what the ASV richness looked like per sample.
In response to the points in this comment:
1. The core was defined from the Venn diagram; we can ensure clarity on this in the methods.
2. Yes, agreed that the findings of the study suggest that "assemblage" is more appropriate than "community"; we will change text as appropriate through the manuscript.
3. Our hypothesis going into the work was that we would see microbial communities that vary in their composition with movement downstream, following downstream change in water chemistry and

other environmental parameters. The presence of this core suite of ASVs is one of the lines of evidence that suggests that mass effects are much more important within this connected system.

4. We can add information on read count and ASV per sample as a table in the Appendix. We provide some information on the composition of the core community in the text (see comment below, and L344), but we can also add a stacked bar plot to the Appendix.

Line 344: the 10 most abundant taxa belonged to 8 different families? This seems improbable. How are you defining taxa here?
Yes, this is correct: the top 10 ASVs belong to these families. We will switch "taxa" for "ASVs" in the text, to ensure clarity. We chose the family level here because we could identify all ASVs to family level, but only some to genus level.

Line 348: I would still like to see the number of ASVs
Please see response above. We can provide this as a table in the Appendix.

Line 350: This is really only reflecting differences between the two rivers, since Bow was excluded. However, if rivers are distinct from each other, and differ by year, should samples be merged? My feeling is that this would be a major reason that the separate effects of individual streams and years needs to be accounted for.
We will add another reminder to the reader that this statement does not include the Bow River.
As described above, we will move forward with a mixed effects model for the chemistry data. At this point in the text, we are referring to the outputs of an NMDS analysis, which is well suited to exploring these types of spatial and temporal differences within the context of microbial community analyses, particularly with respect towards our research objectives within the framework of nearness to respective glaciers and season.

Line 354: To see if environmental variables explain variability is poor justification for conducting an analysis. Please expand on this. Also, how was the % variance adjusted?
The use of RDA to explore how species composition is structured across environmental gradients is quite common in ecology. To ensure our meaning is clear, we will re-word slightly as (or, would welcome further clarification, too):
*We used RDA to assess how ASV composition varied across environmental gradients, to better understand the role of environment in structuring ASV assemblages at each of our sites.*
The suggested addition of hypotheses at the end of the Introduction may help with this too, by making the reasoning a bit more self-evident.
Percent variance was adjusted using the "RsquareAdj" function in the *vegan* package of R. This function uses a permutation method (default 1000 permutations) from Peres-Neto et al. (2006). We can add this detail to the text.

line 375: I would rather see hypotheses led comparisons rather than throwing everything at it and seeing what sticks...it really makes a difference for the reader in terms of focusing on particular results. Also, Im not sure that these are 100% independent datapoints (they have spatial and temporal autocorrelation), yet there comparisons are assuming that they are. I think this really needs to be justified that it is the best approach to show what you want to show. My feeling is that many of the points are already obvious (e.g. headwater indicators being related to temperature. Etc)
Following comments from both reviewers, we will remove this analysis. We agree that it does not add much to the analysis.

line 377: can you say Microbacteriaceae sp? (like family sp.?) To be honest Ive never seen it before, but might be more clear to say its a species from the family Microbacteriaceae
Thanks for catching this - we will edit as suggested.

line 382: speaking of autotrophs, were you able to quantify stream turbidity or otherwise estimate algal biomass? Could help with some of the interpretations….
Unfortunately we don't have these data, no.

Line 426: why is this paradoxical?
In most environments, microbial-origin OM would be expected to be ~modern, so this is what we were trying to get at here.

Line 430: I interpreted this slightly differently…..early in the season, the subglacial flowpaths are inefficient and poorly formed, whereas in the peak melt there is an efficient subglacial channel where most of the supraglacial melt is routed….. I think this needs to just be re-worded to make it more precise…..also, the lack of variability in this study might be due to an inadequate sample size from baseflow conditions?
We can specify that "different OM sources" refers to subglacial sources at this point in the manuscript. It is true that we had many more "melt season" samples than from outside this season, but the number of pre- and post-melt sample points (melt n = 157, pre- and post-melt n = 49) should still be enough to pick up differences in composition.

line 436: Yes, I am wondering how relevant some of these papers on Greenland are for these smaller glaciers…..not that they are bad to cite, but maybe just good to include some smaller ones as well
We can add a reference to Spencer et al. (2014), who similarly found that protein-like fluorescence increased from pre-melt to melt, for the Marshall Glacier (Alaska).

line 468: although there are a lot of other photoautotrophs in the streams besides cyanobacteria, such as Hydrurus, that wouldnt show up in the 16S
Yes, we certainly agree. Unfortunately we don't have 18S data, so use the presence of cyanobacteria, at least, to reinforce the point we're making.

line 490: I think priming in streams is a bit of a debate in general
Agreed - we hope that our similar understanding is reflected in our text.

Line 504: Indeed many of these are found in glacier ecosystems elsewhere, but again phyla are very broad categories. Better would be to identify groups at lower taxonomic levels where more relevant comparison can be made.
At this point in the paper, we chose this taxonomic level to enable direct comparison with the papers that were cited. We can add details at the family level, though this section of the text may be re-worked following comments from Reviewer 2.

512: Do you think that increases in alpha diversity are a response to changing environmental conditions or reflecting a greater number of cell sources? Since the mixing and residence time of streams is relatively short, my guess would be that these are not actually communities but assemblages. It is unlikely from my opinion that they are responding to their environment per se, but more that the assemblage is being formed by the inputs from the surround adjacent landscape. I think the Wilhelm paper was primarily about benthic communities.
Yes - agreed, and we will articulate this better in the paper; while our initial hypothesis was that microbial communities would respond to environmental gradients, it is clear from our results that this is not

occurring (or, at least not to a great extent), and that downstream, an increased variety of hydrologic inputs enables greater assemblage diversity.
Considering the future, one conclusion, then, would be that the relative importance of these different inputs will change, as will their geographic location (e.g., upward migration of forests).
As articulated above, we will switch our terminology from "community" to "assemblage" as appropriate.

Line 522: what are the various lines of evidence? I think you should specifically give the argument….if these are the results from the RDA, keep in mind that this is correlative and not necessarily indicating causation….in this case that organisms prefer or are repelled by a given parameter
Here, we were specifically referring to the RDA and indicator species analyses that are discussed in the few sentences that follow. We can edit the text slightly for clarity, and will also edit to acknowledge the correlative nature of RDA, and following changes to some of the analyses associated with the indicator species outputs (see further comments below).

Line 552: I know that it can include any rank, but when I think of the word 'taxa', I am generally thinking of something at the species or ASV level. To me its really weird to refer to these much higher order taxonomies as 'taxa' when you could instead say 'phyla' 'orders' or 'classes'. I also wonder how much biological sense it makes…..For some of these its almost like referring to 'insects' or 'mammals'.
Throughout the manuscript, we will switch our terminology from "taxa" to ensure specificity. At this point in the text, we will clarify that we are describing family level or lower. At other points in the text (as described elsewhere), we can add this level of taxonomic detail.

Line 565: I think a big difference is that lakes and soils are mostly stationary, while streams by definition are in motion….much more likely to get mass effects in the water column that way, and very difficult for communities to develop
Yes, of course - this is the point we are trying to convey at this point in the manuscript, and can edit to ensure our meaning is clear.

Figure 1: Just like for the binning of longitudinal distances, the binning of the melt period also seems a bit arbitrary. For example, there is only one sampling event that seems clearly to be during a baseflow period. Also, the post melt sampling point in 2020 is associated with the third highest discharge in that given month…..why would this not be associated with the 'melt season'? Furthermore, the sampling of pre/post season samples is really limited in comparison with the melt season samples. Also, how different is post melt to premelt…...could these just be lumped together? What is the rationale to keep them separate (like do you expect distinct patterns in the post melt vs pre melt?)? On the other hand, some of the pre-post melt samples seem that they could very easily belong to the "melt" season, yet are separated by seemingly arbitrary dotted lines in the hydrograph, as the dotted lines appear to be in a different position in 2021 than in 2019 and 2020. Thus, it seems like almost all the samples are taken during the 'melt' period based on the hydrographs. Might it make more sense to derive continuous hydrological variables rather than try to make three categories? Potentially here also continuous variables may also help with interpreting differences in the data, as I can imagine that peaks and troughs within the peak melt season are likely to also have important differences just like at different times of the year? I have to admit, I don't have a better idea, but Im also not convinced that the current strategy is the best one
Thanks for this comment. A few specific responses and points of note:
- There are some sampling points that are not on Figure 1, because they are outside of the hydrograph as measured by the Water Survey of Canada. This is an error in data visualization on our part; we'll modify the figure to show these (December 2019, and January 2021).
- This comment about the post-melt 2020 sample is reasonable. This sampling period followed a late warm period, after which temperatures fell rapidly.

- It's true that we could use discharge at this station (which is ~ 2 km downstream of the terminus of Sunwapta glacier) as an input to our mixed effects model, but that misses that - as the reviewer points out above - we are sampling a series of rivers from three (albeit proximate) glaciers (two icefields). Thus, we prefer the slightly less specific binning approach to acknowledge that there may be some difference in peaks / troughs in flow between glacier outflows, but that the overall hydrologic season "bins" should be reasonably consistent at these proximate locations. Certainly, we didn't want to neglect flow altogether in our analyses.
- The dotted lines aren't arbitrary; they are determined by the 1 m3 cutoff, and so do vary in timing by year. We can modify the plotting of the figure slightly so that this doesn't cause confusion or otherwise become distracting (perhaps, just using different symbol shapes for the three melt periods).

Figure 2: In the box and whisker plot, there is a big white gap where the samples are missing from the COVID period. While I think it is good to be transparent about missing data, it also seems weird to have a big empty spot in the middle of the figure. Would it be possible to just cut this portion out? Also, what happened to the post-melt samples…..were they combined with premelt samples?
Figure 2 will change with the move to a mixed effects model, and thus changes in plotting.

Figure 3: Was deuterium excess different among the three streams as a function of distance? I am just wondering how reasonable it is to lump the sites together in the analyses and figures. Also, while I appreciated that the season was considered in the graphing, it is really hard to pick out any seasonal patterns in these figures based on season, given that the shapes all kinda blur together at some point, and most of the trend seems to be longitudinal.
See figure S1 for trends in d-excess with distance downstream. With this figure, we are illustrating how DOC and DOM composition vary with putative water source. Early tests of a mixed effects model show d-exess to vary significantly by distance (as a continuous variable) and season, with year and river controlled for as a random effect.

**Reviewer 2:**

This study tests how the loss of glaciers will change the composition of organic matter in downstream waters and subsequently microbial community structure. The research question is important given the rapid rate at which glaciers are being lost globally, so should be of widespread interest. The authors, somewhat unsurprisingly, discover clear shifts in the composition of organic matter along the river networks, though the magnitude of the effects on microbial community structure are small. The latter though is somewhat concerning as a negative result and begs the question whether the "right" independent and dependent variables were measured. Nonetheless, I think the paper is well put together.

The technical approaches are sound and well explained, especially the field sampling. However, the data are not statistically independent. There is spatial and temporal autocorrelation in the sampling design, and I do not think that the three-way ANOVAs consider these effects. For example, when estimating responses across distance bins, bins can show more similar values just because they are closer together in space and that needs to be accounted for in the statistical models. Although one could argue that including distance as a fixed factor would enable two closely related bins to have similar values the important point is that we can't disentangle if that effect is purely because of distance or autocorrelation. In other words, assume headwaters and near sites have similar values – is that because each site downstream is similar to its nearest upstream site (spatial autocorrelation) or because there is something special about those distance bins? The same arguments could be made for year and hydrological periods. While the breadth of field sampling is impressive, it is quite complex statistically to analyse something of this nature correctly.

Thank you for these comments. As described above, we will switch from an ANOVA approach to a mixed effects model.

Important clarifications are also required for the microbial analysis. First, were there negative and positive controls in the sequencing? These controls are particularly important given the finding that the same taxa seem to dominate the composition. There should be more evidence given to rule out that lab/field contamination could be a source for the homogenization. Second, what was the read depth and how much did it vary across samples, i.e. are normalization/rarefaction techniques required? Please add this information.

**Controls:** A field blank served a negative control through extraction, amplification, and sequencing. We used this blank to remove ASVs that were likely due to contamination from our samples using the prevalence method with a threshold of 0.5. We did include a negative control in our library preparation steps (just MM, with sterile water subbing in for a sample) to assess contaminating events during the library preparation steps. If we did not see a band following gel electrophoresis, then we proceeded with pooling and subsequent sequencing steps. Sequencing this negative has not historically provided us with enough information to further curate our dataset, and so we did not sequence the negative control. We did not include a positive control because a positive control, for these purposes, aids in determining read coverage for major taxa identified. We have used *E. coli* to spike into some samples as a positive control in the past, and have recovered expected quantities from sequenced data. As such, we did not include it here, especially because we have spiked in PhiX quite heavily during the sequencing run (~50%), to increase the diversity of the oligotrophic pool, so the reads we do obtain will correspond with mostly the most abundant microbial taxa in the sample, as a function of this and the read coverage of the MiSeq. We can add this caveat to the work.

**Read depth:** As described, above, there was a large range in read depth, ranging to ~300,000 reads. Samples below 5,000 were removed because these samples had not reached a plateau (see Figure 2, above), indicating they are not a representative sample.

**Normalization:** We Hellinger transformed our data as it gave us the most normally distributed data. We also ran the analysis on each of rarefied, Hellinger transformed and un-transformed data and did not see large differences in the results.

In all cases, we will add this information to the methods of the paper.

[Figure]

**Figure 3:** *ASV count frequency for rarefied (L panel) and Hellinger-transformed (R panel) data.*

**Specific comments**:

Line 29:  I don't follow how this is necessarily "complex" given the previous conclusions of a core set of species that overwhelm (mass effect) environmental gradients, suggesting it is a simple predictable outcome.
As pointed out by reviewer 1, "complex ecosystem responses" is also a little bit vague. We will reword this sentence to describe changes in organic matter and microbial source pools, and how this may affect DOM composition and microbial assemblages across our study sites.

Line 66:  shift"s"
We will edit as suggested.

Line 90:  The term "mass effects" is jargon and should be defined on first use, especially for biogeochemists that may be less familiar with this "ecological" term.
We will define on first use.

Line 153:  What volume of water was sampled for the microbes?
We collected 2L of water for each sample, and processed through the sterivex filter until the filter clogged.

Line 155:  Not sure I follow the logic for why the microbes would change in the Bow River samples but not the carbon...  They're linked…that's the argument of this entire paper.
Carbon was filtered streamside, and so we didn't have the same concerns for these samples. We will change L142 to read "always filtered streamside" to ensure clarity on this point.

Line 265:  Please can you explain why this is necessary for this comparison.
This isn't strictly necessary, but was done for visualization purposes. We will clarify our rationale in the manuscript (and see also our response to reviewer 1, above).

Line 267:  Beta-diversity is calculated from the Bray Curtis index not the NMDS.  NMDS is simply a visualization technique.
Agreed - we will modify terminology on this point.

Line 269:  How was the perMANOVA performed?  And how were the clusters identified?
The perMANOVA was performed in the R environment, using the "adonis" function from package *Vegan*, set at 999 permutations, to assess the significance between groups (based on distance from glacier). This test was conducted on a Bray Curtis dissimilarity matrix, constructed from our original microbial community dataset.  Clusters followed the "headwater", "near" and "far" bins, which were constructed *a priori*. perMANOVA was conducted on these bins. We will add some details on the analysis at this point in the text, and will also flesh out our *a priori* choice of bins, as described in the response to reviewer 1.

Line 270:  Why have you performed the RDA?  Please explain the biogeochemical question you are trying to test.
See also our response to Reviewer 1, above. We used RDA to assess the degree to which microbial assemblages varied across gradients of the environmental parameters that we measured, to help to explore whether assemblages were shaped by environmental drivers. We will ensure that this reasoning is articulated in the text.

Line 287 and throughout: The test statistics associated with the p-values must be reported to be reproducible. I presume here you should have some F statistic from the ANOVAs with some degrees of freedom?

We had kept the F statistics and degrees of freedom in the supplemental tables, to avoid cluttering the main text. However, with the move to a mixed effects model this text will change.

Line 335: I don't follow what is meant by "passive overlay".

We will clarify this in the text, for others who may be unfamiliar. We are referring to a vector that is overlain on the PCA after the ordination has been conducted, such that it is shown in ordination space, but does not affect the structure of the ordination. In this case, the intent of the PCA was to explore variation in DOM composition across sites (and secondarily, "seasons"). d-excess was then added after the fact, to explore its association with the data in PCA space.

Line 341: Aren't these 10 phyla dominant in most rivers? It would be useful to contextualise these results, such as through comparison with the Earth Microbiome Project.

Yes, it is true that these are typically the most dominant phyla found in most rivers. We contextualize our findings with other systems in the discussion (Lines 500-510), and will certainly add such comparisons using lower taxonomic resolution, following this comment and the comments from Reviewer 1.
A detailed comparison to the EMP project is outside the scope of this manuscript, though perhaps something that could be done with this data in the future. While we very much agree with the reviewer's comments about the usefulness of lower taxonomic resolution comparisons and contextualization, we note that the reason that much of our text is at the phyla level is because of our wish to make comparisons with previously published literature, which typically also presents results at this taxonomic resolution.

Line 356: Please cite evidence showing that these parameters were highly inter-correlated.

We can add the r value that was used to make this assessment to the text.

Line 374: There are no correlation statistics given anywhere to support this claim, i.e. of a trend in the clouds shown in Fig. 10.

We can add statistical information at this point in the text, and in the caption for Figure 10.

Line 419: But is there enough of this material in a mass-balance sense to matter?

Yes, this is a fair point, and we can add a few words to address this caveat. Note that we get into this more specifically for the DOM pool at the end of section 4.4.

Line 474: A mixing model would really be the way to get at this question and the Discussion could at the very least point to its utility.

Yes, agreed. We shied away from a mixing model, given the mathematical requirement that the number of "endmembers" be no greater than (n+1) the number of tracers; at least to enable a robust model.
However, we can certainly point to the utility of this type of approach in the Discussion.

Line 498: I don't think this paper tests this relationship as it cannot disentangle create from consumption of OM. The rest of this paragraph also says little about this question and just reviews the composition of bacterial families.

Yes, we agree that disentangling this relationship is beyond the scope of the paper, and was not - in fact - one of our objectives. This issue here may actually be in the sentence that follows (*"To further explore this relationship …"*). We will edit to ensure we're not being unclear on our intended scope.

Line 524:  I think this statement overstretches.  These were statistically significant but explained very little variation.  A total of like 9% all together, so how important was each variable?  I think the discussion that follows on lines 538 is much fairer.

Thanks for this comment.  Our intent here was to: (1) not discard the importance of environment altogether (see for example the reviewer's comment at L544), but also (2) provide some bridge text that moves to L538, where we discuss the fact that mass effects are much more important for regulating the composition of these assemblages. We will restructure to ensure clarity on our meaning.

Line 525:  Again, what is the biological significance and effect size?

We can add statistical outputs at this point in the text.

Line 544:  Or we're not measuring the "right" drivers.

Yes, indeed. Though, as review 1 points out, our suite of environmental drivers was fairly extensive.

Line 546:  Again, lack of strong "control" given the variables that were measured.  That's the problem with a negative result – is it the truth or the study?

True, there could be some environmental control that we are not assessing. We can rephrase here, by rewording as "this lack of strong environmental control *by the variables that we measured*", and otherwise softening the language.

Line 591:  Please cite some evidence to support this statement.  It does depend on the functional redundancy within these communities.

We can soften or otherwise edit this statement, given that - as the reviewer notes - it does depend on functional redundancy.

Figure 4:  I don't follow which end member the grey box corresponds with.  It looks like only part of the grey box corresponds with different end members.

The grey box corresponds with end-member 4. We can articulate how end-members are visualized (arrows, grey box, pink box) in the caption, to ensure clarity.

Figure 6:  I think you should add to the caption that the percentages along the axes are explained variance.

This can be added to the caption.

Figure 8:  I don't follow why there are two circles if there are three groups for distance and season – which of these variables correspond to the circles and why is being omitted?  I think you are looking at distance and grouping near and headwater together but the figure should be self-contained with its caption.

This is correct - in this figure the near and headwater sites are grouped because perMANOVA found that they were not distinct, but that both differed from the far sites.  Plots showing other comparisons (between rivers and years) and a table with the perMANOVA results are in the supplement.  We will add detail to the caption to ensure that the figure is self-contained.

Figure 9:  I find it confusing to have three values for the variance explained, none of which match each other.  So the first two axes of the RDA explain 14%, all axes together explain 25%, and all axes when adjusted for the number of predictors explain 9%.  Is that correct?  Is it possible to focus on one number and just be more forthright that none of the environmental predictors do a particularly good job here?  As for the crosses in the centre, how come the near melt sites in the bottom-left are so far away from the taxa?  Are there no unique taxa associated with them?  It would be informative to see the indicator species labelled on here.

Expressing the variance explained is something that the co-authors on the paper went back and forth on a fair bit. We wanted to ensure that we were representing our outputs correctly (i.e., not overstating our results by not presenting the corrected $R^2$), but also wanted to ensure clarity. We landed on the approach in this figure, which seems like it is a bit unclear to an outside reader. We will move the $R^2$ values in the bottom right of the figure (adjusted and all axes) to the caption, where we can provide a bit more context on meaning.

The species scores' clustering near the center of the dbRDA reflects the overall poor explanatory power of the ordination. We did try different plottings (including adding the indicators to this plot), and didn't find it added to our interpretation.

Figure 11: There are ca. 85 correlations here. Are you not worried about false positives, especially given that some of these Rho values look small? Also, I don't follow how the Microbactericae - temperature correlation can be more statistically significant than the one between Beggiatoceae and temperature but have a smaller absolute rho value. I haven't checked all the other columns for similar problems.

Following this comment and the comment from reviewer 1, we have decided to remove this figure. However, we note that the correlations were directly output from the statistical package used to undertake the analyses, and so this 'offset' seems due to differences in sample size.

**Literature cited:**

Cui, X., Bianchi, T. S., Jaeger, J. M., and Smith, R. W. (2016). Biospheric and petrogenic organic carbon flux along southeast Alaska. *Earth and Planetary Science Letters.* 452, 238-246.

Peres-Neto, P.R., Legendre, P., Dray, S. and Borcard, D. (2006). Variation partitioning o species data matrices: Estimation and comparison of fractions. *Ecology*. 87, 2614-2625.

Serbu, J. A., St. Louis, V. L., Emmerton, C. A., Tank, S. E., Criscitiello, A. S., Silins, U., et al. (2024). A comprehensive biogeochemical assessment of climate-threatened glacial river headwaters on the eastern slopes of the Canadian Rocky Mountains. *Journal of Geophysical Research: Biogeosciences.* 129, e2023JG007745.

Spencer, R.G.M., Vermilyea, A., Fellman, J., Raymond, P., Stubbins, A., Scott, D., and Hood, E. 2014. Seasonal variability of organic matter composition in an Alaskan glacier outflow: insights into glacier carbon sources. *Environmental Research Letters,* 9**,** 055005

---

## Author Response (AR1)

We would like to thank the two reviewers and the Associate Editor for the time that they have taken with our manuscript. We apologize for the relatively lengthy period that it has taken us to return a revised manuscript for re-assessment. Shortly after this manuscript's initial submission, one of the two supervising authors, Dr. Maya Bhatia, died unexpectedly in a tragic field accident (see the compiled tribute to Maya on Wikipedia, here). Although this has been incredibly challenging for our co-authorship team, it has also been extremely important to us that we see the paper through to publication. For various members of the team, Maya was our supervisor, our close colleague, and our friend.

We have worked hard to address the reviewer comments, and have considered each of the comments quite carefully. Most of the suggested changes have been implemented, including moving to a mixed effects model to analyze the chemistry data contained within the paper. In cases where we did not implement a suggested change, we have provided a detailed rationale as to why in our responses below. Although the decisions made regarding revisions were purely scientifically based, we are hopeful that the reviewers will be understanding that varied approaches can be suitable for analyzing a given dataset, and also be sensitive to the emotional difficulties associated with re-working this particular manuscript, following Maya's passing.

Thank you again for the helpful and thought-provoking comments on our paper, and with best regards,

Suzanne Tank
on behalf of all co-authors

**Reviewer 1:**

In this manuscript, the authors have addressed an important question, effectively how glacier fed streams change longitudinally, which I fully believe is in the interest of the Biogeosciences readership. The data themselves are very interesting (with some nice longitudinal patterns), as the analytes chosen are good proxies for glacier processes and for relevant biogeochemical cycles. The paper is overall also quite well written, and the figures well illustrated. However, in carefully reading through the methods and results sections, I have three major comments: 1) The statistical analysis can be improved. The issue to me is that there are three streams, sampled at different points in the hydrograph (premelt, melt, postmelt), at 3-4 different transects, over three years. This is already a lot of categories for the dataset size which makes interpretation tough. Also problematic is that the categories are also not neatly defined: the 'longitudinal' categories encompass quite some different distances, and there is uneven replication and temporal representation (mostly in melt versus pre- or post- melt, the distinction between each seems arbitrary). Another issue is it seems that the data for the three streams are lumped together in the analyses. Can this really be justified…..wouldn't it be better to analyze each glacier stream individually? As a result, I think that there could be quite a problem with spatial and temporal autocorrelation, and I think data independence is an assumption with the utilized analyses. I have read through this paper several times now, and I must admit, I don't have an obviously better solution given the sampling design, but one possibility could perhaps be some mixed models (ex GLM or GAMM etc) that account for the different categories, as well as the possibility of making some of the categorical variables continuous (i.e. distance from the source glaciers) to improve subsequent interpretations. 2) Another general issue is the justification for why these things are done in the first place….what do the authors hypothesize, particularly with regard to each of these tested categories? The stated rationale of seeing how they change with distance downstream can be much elaborated upon. 3) Finally, I was a bit disappointed in the lack of information regarding the microbiological analyses and results (in more detail below), which are really necessary to assess the quality

of the data and interpret the results. Therefore, I challenge the authors address these three major points in their revision. Some more detailed points follow by line number.

We would like to thank the reviewer for the thoughtful review of our manuscript. In relation to the specific points raised above:

(1) Thank you for this statistical suggestion. We had considered a mixed effects model at the outset of manuscript preparation, but chose to move forward with the "binning" approach, mostly for data visualization reasons. We now use a mixed model for our analyses, using a model that includes distance and melt season as fixed effects, and year and river as random effects, and can confirm it is workable, and does not change the major conclusions of the manuscript.

(2) We have added specific hypotheses at the end of the introduction, as also suggested in the detailed comments below.

(3) We have added more details on the microbial analyses, as further laid out in the specific comments below.

Line 13: be careful here…..not sure that this OM is necessarily structuring water column communities if they were just exported there, and especially given the result that mass effects seems more prevalent than environmental filtering

We have rephrased the wording on this point, now as: *and its _possible_ role in structuring microbial assemblages has yet to be characterized in the Canadian Rockies.*
At this point in the manuscript (in the abstract, stating that this has not been explored) we felt the overall wording was ok.

line 18: microbial communities inhabiting what? There are a lot of habitats in streams, and I think this detail would be of interest to those reading the abstract. As I have already read the paper, I feel like these are probably not communities as all, but microbial assemblages in transport. I think this should be clearly stated here, and I make further remarks on this below

Samples for microbial community analyses were collected as described in the previous sentence. We do not expand in this sentence to allow for brevity in the abstract.

line 21: although I never really read any names of the putatively chemolithoautrophs or cold adapted taxa….mostly just phyla and other high-order taxonomic names

This point in the abstract specifically refers to the indicator species. We first introduce the indicator species in the results (Sn 3.2.1), and then provide details on their characteristics in the Discussion (section 4.5). Hopefully this will stand out a bit better with the edits that have been undertaken in the Discussion. Given the need for brevity in the abstract, we will refrain from adding more specificity here.

line 22: while I don't dispute the role of glaciers in 'seeding' headwaters, im not sure the data really show this…..you don't have glacier endmembers, and you cannot prove that the microbes were actually alive or living there, so there is not really a 'smoking gun'….just some precision may be needed in the language

Our furthest upstream sampling point was ~300 m from the glacier termini, so the vast majority of water (certainly during the melt season) was glacially-derived. We certainly agree, however, that mass effects appeared to dominate; we have adjusted our wording slightly here to ensure precision.

line 23: probably all could be argued to be indicators of water source

True! This qualifier was added at this point in the text because deuterium excess (i.e., the isotopic composition of water) is not really an "environmental condition", per se. We have rephrased as: *"…physical and chemical conditions (including water temperature; POC concentration; protein-like DOM; and deuterium excess, which we use as an additional indicator of water source) …"*

line 29: 'complex ecosystem responses' sounds pretty vague….could you make any more specific predictions?
Agreed. We have modified the text at this point to refer to loss of glacial seeding, and concomitant "terrestrialization" of organic matter chemistry that may also cause change in microbial community composition.

Line 43: 'supraglacial' and 'subglacial' would be related to glacier tops and bottoms, respectively. Marginal channels are not found on top or beneath the glacier (unless I am misunderstanding something).
We have rephrased this sentence as follows:
*As glaciers melt, the formation of supra- and subglacial channels can cause meltwater to be routed on top of, through, and/or underneath, glaciers*.

Line 49: This process also helps to explain a lot of the differences in your analyzed variables, which should probably be discussed to justify their collection
Agreed. We have altered this sentence to refer to both hydrology and biogeochemistry, with an added biogeochemical reference.

line 56: the POC is from crushed rock? Couldn't it also be from the overidden vegetation?
We have switched our wording here to read "*can be predominantly derived from*", and add a reference to Cai et al. (2016) that quantifies this effect. At these Alaskan sites, glacial-origin POC was found to be predominantly petrogenic in origin, although certainly overridden vegetation would also contribute to this pool.

Line 64: agreed that these are important variables for benthic communities, but do you think the same is true for the organisms in the water column?
Yes, we do think so - we would argue that light penetration (particularly in sediment-laden streams) and flow (associated with sediment, but also generation time) are important for water column communities, as well as those that are associated with the benthos.

Line 73: this seems to be a specific number…...from my experience, glacier surfaces are remarkably heterogeneous, which may be good to note.
Yes, fair comment – we have broadened the scope of this statement, which is specific to one publication.

line 78: although there is only microbial data for the headwaters and far reaches, and none from Bow
It's true that we only present data from headwater (300 m downstream of glacier), near (3-7 km downstream), and far (50-100 km downstream) sites, which is a decision that we made for data visualization purposes - it's such a rich dataset!  Reasons for excluding the Bow microbial data are described elsewhere. Re-reading this sentence, we feel it is still "true" to the data analysis that we did, and the dataset we present.  We will swap "rivers" for "reaches" to ensure there is no confusion on the scope of the work.

line 81: how specifically can this work inform us about how glacier loss will impact microbial diversity?
At this point in the manuscript, via either: (1) loss of source assemblages, or (2) changes to "environmental" condition (water chemistry, temperature, etc.) that drive changes in microbial assemblage composition. Hopefully with surrounding edits, this will be clarified.

line 84: this is repeated from the first paragraph, line 34 I think….
Thanks for catching this; we have deleted to avoid redundancy.

Line 90 to line 93: While this doesn't bother me, it seems a bit strange to give the results of the paper in this last paragraph…..this could potentially be skipped. What would be nice to include, however, would be some specific hypotheses that could be tested. What exactly did you predict would happen to the OM and why? How should microbial communities change with space/time and why?
We are happy to provide hypotheses at this point in the paper.

At the outset, our hypothesis was that we would see clear variation in chemistry with movement from glaciers to downstream reaches, and that this would structure microbial assemblages, thus causing these to also vary along this gradient. Setting this hypothesis up at the outset also helps us to reinforce some of the key take-aways of the paper (i.e., evidence for mass effects) and thus strengthen the paper overall - thanks for the suggestion.

General comment for introduction: The introduction is written nicely, but many of the papers used as citations are from work on the Greenland Ice Sheet, which is really an enormous piece of ice with its own very special characteristics. I know nothing about the glaciers which were the basis for this study, but I encourage the authors to evaluate whether or not these citation are applicable in the context of their study system, and if some papers dealing with smaller glacier might be more relevant to cite in some cases.
Agreed that the Greenland Ice Sheet is very different from the mountain glacier system that we are studying. In the Introduction, we draw from studies conducted in Greenland, but also from glaciers in the Alps and Alaska. There has actually been very little work on carbon biogeochemistry specific to Rocky Mountain glaciers, which was part of the impetus for our study! We prefer to keep this diversity of references, because they show the consistency of what one might expect regarding glacially-sourced organic matter composition, for example, and thus helped to set up the initial hypotheses for the study.

Line 96: I see that there is a hydrograph for one of the streams, but how do the discharges of the other two streams compare with this one? Could be much different given that different size of the icefields mentioned here, yet there is no way to tell. This is an important consideration given that all three rivers were lumped together in the analyses, though my feeling is that the effect of 'stream identity' should be taken into consideration.
Please see below (response to comment at L118) about availability of hydrographs during our study year, and consistency of discharge across sites.

Following the comments of both reviewers, we have changed our analytical approach to more specifically account for river identity (see more detailed comments above on our mixed effects model).

Line 103: be careful with the word "evolve", since it has quite a specific meaning in the biological sciences
We have switched "evolve" to "transitioned" at this point in the text.

Line 105: These glacier distance binnings seem arbitrary to me. Is there some ecological reason that we could expect changes with distances of these magnitudes, or are they indeed arbitrary? A concern for me is that the difference in the far sites (100-40=60km) is actually greater than the distance of the first three other categories (0 to 35 km), as well as 1/3 of the 'far' category itself. I guess my question is if these categories make ecological sense given this, or would it make more sense to use distances from the glacier as a continuous variable, in which case the 'realistic nature' of these categories would no longer matter?
These distance bins were selected *a priori* to reflect the substantial difference in watershed composition between bins, as follows:
- The headwater site was immediately (a few hundred meters) downstream of the glacier termini, and thus received almost all melt-season runoff from glacial sources. Within its watershed, this site had 0% forest cover, 50.7% coverage by snow and ice, and 48.8% coverage by rock and rubble

- Near sites were 2 - 5.6 km downsteam of glacier termini, and above the point of substantial forest establishment in the catchment, but with a large enough catchment area that source waters would be from variable sources. These sites all had less than 5% forest cover.
- Far sites (greater than 50 km downstream) were at sites with significant forest (and thus soil) development adjacent to the sampling site, and with 25% forest cover within the broader watershed (i.e., including upstream, mountainous, reaches).

We added these details to the current Table S1, and flesh out this *a priori* rationale at this point in the text.

However, given the similar comments from both reviewers, we have switch to a mixed effects model for our chemistry analyses (i.e., move away from the ANOVA approach), which includes distance as a continuous variable. We agree with the reviewers that this is a more suitable statistical approach, though it does not change the overall conclusions of the paper. This has meant changing most box-plots figures to scatter plots with distance plotted as the independent variable.

For the microbial analysis, we continued to use these distance bins (e.g., as an input to perMANOVA in the NMDS, the indicator species analysis, and the Venn diagram and associated analyses); we had previously experimented with other approaches and data visualizations, and looked at this again – in detail – for the manuscript revisions. In the end, we have decided that these *a priori* bins are the best way to assess this very rich dataset.

Line 108: I think "stream" works, but its then a bit weird then that you used 'river' in the title
For clarity, we have rephrased the title as follows:
*Shifts in organic matter character and microbial assemblages from glacial headwaters to downstream reaches in the Canadian Rocky Mountains*

Line 113: how many of these samples were taken in December and January? There are not a whole lot of winter glacier fed stream data, so these could be of extra interest…. Where they different than the others? Unfortunately they are buried within the rest of the other data and its not possible to see them. They also don't seem plotted on the hydrographs. Thus, despite reporting them here, they seem invisible in the paper.
Thanks for pointing out that these samples seem to disappear in the manuscript. We sampled in December of 2019 and January of 2021 (i.e., during under ice periods after the first and second open water sampling years). Of our sites, we were only able to access three in 2019 and four in 2021, as a result of poor site accessibility and overall safety considerations. We have modified figure 1 to add these data points beside the hydrograph: they were not in the initial figure because they fall outside of the recorded hydrograph. Following these comments, we have also added the winter samples to the mixed effects model analyses.

Line 115: though the sample coverage over these three periods is very uneven, and most samples are from the melt period. Should put some sample numbers here to give the reader an idea of how many samples were collected at each period. Also, do you have a feel for how well the Athabasca Glacier hydrograph corresponds to the hydrographs of the other two streams?
We have added the number of sampling campaigns in each period at this point in the manuscript.
The second question is a challenging one, because the Sunwapta outflow is the only one that is gauged. However, the three glaciers are in quite close proximity to each other and so experience relatively similar climate (i.e., presumed start to melt season). And, we know from our own field observation that the onset of more rapid flow (i.e., associated with the summer melt season) was similar across our sites. The outflow of the Bow Glacier was gauged in 1973 and 1975 (since discontinued); of these two years, 1975 had a much longer (but still not full) melt-season record. We provide a figure that superimposes the Bow and Athabasca

glacier outflows below, illustrating synchronous pulses in streamflow that presumably reflect relatively homogenous temperature and precipitation patterns within this region.

[Figure]

*Figure 1: Measured discharge of the Bow Glacier Outflow (Station number 05BA009), and Sunwapta River at Athabasca River (station number 07AA007) during the open water season of 1975.*

Line 118: instead of arbitrarily using 1 m3, couldn't you use your hydrological/chemical data you collected to determine if the subglacial channels have opened or not? By using arbitrary cutoffs as you have here, there is a good chance that you will get results from your analyses without the stated ecological relevance. Also, what kinds of differences might you expect to appear in your data as a function of season? These hypotheses are not clearly identified….

The 1 $m^3$ threshold was chosen following close inspection of the Sunwapta River at Athabasca Glacier hydrograph (i.e., the discharge monitoring station most immediately downstream of this glacier). Although the 1 $m^3$ may appear to be arbitrary, the cutoff was chosen because it represented the inflection point characteristic of meltwater contributions to flow. We rephrased in the manuscript to clarify this point; thanks for pointing out that we had neglected to do so.

Line 125: thus sample year should probably also be accounted for in an eventual model

See above for changes to the statistical approach, and our planned switch to a mixed effects model for this component of our statistical analysis. This approach now includes sample year (though we note that our original approach via ANOVA did also include year as a factor).

Line 126: would be nice to know how many samples were collected from each of these categories. I know that some of this information is in Table S1 (i.e. sites as a function of distance), but it would be really good to have some of these numbers here too. From looking at the plots, its seems that the pre-melt and post-melt periods are vastly undersampled compared to melt periods, yet there is no way to really know this except by looking at figures.

We added this information at this point in the text (see above).  It is true that we have many more samples during the melt season;  however, this is by virtue of the fact that we sampled every 3-4 weeks from the onset of streamflow through until September, and then twice (October and under ice) thereafter. Thus, it's really just that the dataset is incredibly rich, particularly during the summer months. We also note that this type of binning (pre-melt; melt season; post-melt) is quite common in the published literature for glacier-associated biogeochemical studies.

Line 133: should probably also put the charges on the trace metals since there are charges on the other ions (also on line 200). Also, why exactly were these measured? There is no hypothesis stated, and the data arent really discussed anywhere else? Same could be said for the major ions for that matter.

We have added charges. We include these data to enable a full assessment of possible environmental controls on microbial community composition (see also our response below).  A more thorough work-up of the trace metal and nutrient data is provided in a recent publication by Serbu et al. (2024), which we now reference directly in our manuscript.

Line 134: I think its common to acid wash bottles for nutrients too, although I realize this wasnt the focus on the work, and the difference in results likely not large. What does the citranox do?
The analyses were run in a CALA (Canadian Association for Laboratory Accreditation)-certified lab, which is - among other analyses - certified for the analysis of nutrients in freshwaters. The lab has tested collecting nutrients using new, unwashed bottles of this specific type, and found that this gave better (lower variability, lower and more consistent blanks) results when compared to reusing (and washing) bottles. Our procedures followed this SOP.

Citranox is a detergent which is phosphate free, and is commonly used to keep labware and equipment clean in many environmental laboratories. It is an extra cleaning step prior to acid washing that does not introduce phosphate contamination, and is able to remove any metal oxides, salts and inorganic residues that may remain in the sampling bottles, without leaving behind any contaminating residues.

Line 150: how much water did you filter for the microbes, and what exactly does 'prepared' mean in terms of the bottle?
We collected 2L for each microbial sample, and filtered "until refusal" of the sterivex filter. In some cases, filters were loaded before the full 2L was filtered. We have added this detail to the text.

Prepared just means cleaned, as described in the previous paragraph. We have switched our wording for clarity.

Line 200: calling dSi a nutrient is contentious. Also, what were the limits of detection for your nutrient analyses?
We prefer to include Si in the list of nutrients because it can of course be limiting in some situations. We have added LOD details to the supplement.

Line 204: could give the nutrient methods numbers here as well
We have added methods numbers to the supplemental table that provides the LOD; as described above, all analyses were conducted at lab that was CALA certified for these analyses.

Line 215: do you have a citation for the modifications of the protocol? Or do you have some data on what kind of improvement you can expect? Just that this might be of interest to others doing this kind of extractions with these kits
We do not have a citation for this modification: this was a modification based on the expertise of Dr. Josh Neufeld (University of Waterloo, Canada), who had advised one of our authors when this protocol was first implemented (2016: much earlier than this study) for similar work that dealt with highly oligotrophic waters. The results from this modification suited our needs (DNA yields were often >1 ng), and we carried forward this technique for this study. For context, there was concern that 5 minutes at such a high temperature may degrade any microbial DNA that had been captured outside cells on the filter, and/or would not be as efficacious in lysing cells uniformly.

line 233 and elsewhere: I think that there should be quite a lot more information on the microbial data. For example, how much water was possible to pass through the sterivex in the end? How much DNA were you able to get from the filters following extraction? What kind of depth did you sequence to/how many reads

per sample? Were they rarefied for the analyses? I think these are pretty important things to report given that these are often difficult habitat types to work with…..

Thanks for this: We have added more information on the microbial data, either at this point in the methods, or early in the Results, and agree that more detail is helpful. To answer specific questions here:

-Details on filtration are in our response above (up to 2L filtered)

-Measured DNA from filters ranged from below detection to 20.8 ng/ul; when DNA concentrations were very low (determined by observing <10ng/uL on Qubit following DNA extraction), we added 5uL of template DNA (default is 2.5uL), along with BSA to coat the sides of the PCR reaction tube to maximize DNA available for PCR, adjusting for total water added to the reaction.

-We had as many as 300,000 reads per sample, but also with large variability; samples with less than 5,000 reads were excluded because they had not reached a plateau, indicating that sequencing might not have captured a good representation of species (see Figure 2, below).

-We chose not to rarefy because of the well-known debate on this topic. However, we did run all microbial analyses with a rarefied dataset, and this did not change our overall results.

[Figure]

*Figure 2:* *Rarefaction curves to illustrate the number of amplicon sequence variants detected as function of sequence depth on individual samples.*

line 235: what do you mean by "low abundance ASVs"? Like….you removed rare ASVs? At what threshold….why??

Here, we were actually referring to ASVs that were present as singletons. We have reworded for clarity.

line 242: not a big deal, but seems that the DOM ab/fluor should go with the DOC methods above rather than the microbial data

Looking at this again, we think it makes more sense to move this text to section 2.4 (Data analysis), since it focuses on data post-processing. We have reconfigured in the revised manuscript.

line 255: why did you choose three way ANOVA rather than keeping these as continuous variables? If you made mixed models or similar, you could control for the effects of stream, season, etc, while also using the data directly as continuous variables rather than making arbitrary categories for downstream distance, discharge, etc
See comment above – we have moved to a mixed-effects model, and have tested to confirm that this is workable for our dataset.

line 260: though I think deuterium can also differ from glacier to glacier…..would be possible that these numbers different between the headwaters of the three streams. Also, why plot this as a passive overlay on the ordination? Should explain this….
Yes, it's possible that d-excess varied between the two glaciers, though this is a well-used metric for water source in glacial and other high elevation studies.

We plotted as a passive overlay because the ordination focused on organic matter composition, and so was confined to these specific parameters. The passive overlay of d-excess is to allow the reader to make some conclusions about water source (acknowledging that this is not perfect …), and how DOM composition varied with this parameter.

Line 265: I think it should still be possible to compare and contrast headwaters and downstream samples with the middle sites included…..not sure I understand this justification….I would probably just include all of the data, no?
The reviewer is correct that including the mid sites does not change the conclusions of the manuscript. However, it does improve our data visualization quite a bit. After iterating on our data presentation a fair amount, we decided to move forward with this presentation, and would prefer to stick with this approach. We do note, though, that all data are included in our dataset upload to NCBI, enabling others to explore these data further.

Line 266: maybe I am old fashioned, but I like to know the actual number of ASVs rather than using shannon for alpha diversity
We specifically chose to calculate and present diversity (i.e., via Shannon Weiner) rather total ASV count because: (1) our focus was on diversity, rather than richness, and (2) total ASV count is of course affected by the amplification process, and so not a "true" estimate of taxa richness, in the way that estimates of richness for other taxonomic groups are understood to be.

line 268: is it necessary to square root transform AND calculate bray curtis distances?
Yes, it is preferable (and common, i.e: Zorz et al., 2019, Laporte et al., 2021…) to do both on a microbial dataset, usually because microbial community data does not represent a normal distribution amongst all samples (large abundance of microbial members in a few samples, but several samples with much smaller abundance), which can be an effect of sequencing depth. This can skew beta diversity analyses, such as what is visualized by the NMDS, if we do not first standardize the data. Transforming the data by Hellinger transformation allows us to standardize the microbial community dataset by assigning lesser weight to samples containing low counts and/or 0 of particular microbial amplicon sequence variant (ASV), which is a common attribute of microbial community data. The application of a bray curtis distance then determines how different the abundances of microbial members are between samples more accurately, versus if we were to simply use a bray curtis dissimilarity matrix without standardization.

Line 269: so you identified clusters on the figure and then tested the clusters with permanova? That sounds a bit self-fulfilling to me…..wouldnt it be better to test hypotheses using the data? Also, there are likely far too many parameters that were included in the dbRDA, and its not clear why most of them were included.

Perhaps you can provide some justification for why some of these are here? For example the trace metals? How many factors were left after the highly correlated ones were omitted?

As described above, we identified clusters *a priori* using distance-based bins (Figures 7-9), and also (Figure S6) by river and year. perMANOVA was then applied to these *a priori* clusters. We have provided more detail on the a priori selection of the distance bins in section 2.1 of the revised manuscript, following on comments from both reviewers.

Similarly, we did not want to exclude any environmental parameters *a priori* in the dbRDA, because of the broad suite of factors that is known to affect microbial assemblages. In this case, we did not feel that we had good *a priori* reasons to exclude environmental parameters, and so included all data that were available.  The poor explanatory power of the RDA perhaps reinforces this decision - if we had run it with just a subset of the environmental data that we had available, we likely would have chosen to 'circle back' to see if we were missing something. We had 12 parameters remaining after highly correlated parameters were removed (following an assessment of variance inflation factors), and additionally used a backward selection approach to assess significant contributors to model fit, rather than including all parameters in the final model.

Line 278: Should probably correct for multiple testing, no? Also, what exactly will testing co-variation between indicator species and environmental parameters tell you? My feeling is that indicators of upstream will just be correlated with things more likely to be characteristic of upstream sites, like low temperature for example

Following comments from both reviewers, we've decided to exclude this figure from the manuscript.

line 280: well…..its the R statistical environment, which uses the R programming language

We have changed "*with the R programming language*" to "*within the R statistical environment*".

Line 287: what kind of test does this p value correspond with? Also, there is likely to be big differences in sample sizes between categories…..were there differences between rivers, and if so, can you justify lumping the data together?

This p-value corresponded to the ANOVA, which included river as a factor. However, this presentation has changed with the move to a mixed effects model.

Line 294: although there wasnt so much pre-melt data for the other years…..

Comparatively, perhaps, but overall we would argue that this is quite a rich data set!

line 297: 'largely non-significant trend'…..probably it was just not significant

This wording has changed with the move to a mixed effects model.

line 319: could be due to the high/low points on the hydrograph during the meltseason, which could be related to greater sub/supraglacial contributions

Thanks for this insightful comment - the more enriched headwater PO14C sample was taken on 23 June 2020 (i.e., second sampling point in 2020), when flows were quite high.  The more depleted samples (across pre-melt and melt) were taken across a variety of flow conditions. We don't see a clear relationship, but this is a very good question.

Line 338: These phyla are literally everywhere, thus making this sentence not very informative. Not that it is necessarily bad to mention these, but I would focus on lower taxonomic levels….

Agreed. We have kept these two introductory sentences that focus on phyla, in part because other, similar, manuscripts have focused their data presentation at this level, and so this information is useful for

comparison purposes. We now include a description of top orders to both the Results and Discussion, in addition to the indicator species analysis that was already present.

Line 343: How was the core defined? There are many many ways to do this, and it would be good to know what were your assumptions….probably in the methods. Also, there are two interesting things that come up with this: First, you call it a community, but im not sure this is an accurate term to apply to these microbes, given that almost all of them are probably being passively transported, and therefore not 'residents'. Also, why do you think there should be a core…..what are your hypotheses here….how is this core maintained? In any case, a core of 1,409 ASVs tells us very little because we have no idea who they were, how many reads were generated per sample, and what the ASV richness looked like per sample.
In response to the points in this comment:
1. The core was defined from the Venn diagram; we have added some more details on this in the methods (section 2.4.3).
2. Yes, agreed that the findings of the study suggest that "assemblage" is more appropriate than "community"; we have changed text as appropriate through the manuscript.
3. Our hypothesis going into the work was that we would see microbial communities that vary in their composition with movement downstream, following downstream change in water chemistry and other environmental parameters. The presence of this core suite of ASVs is one of the lines of evidence that suggests that mass effects are much more important within this connected system.
4. We now include rarefaction curves in the Appendix, and have added information on the overall range of ASVs and read depths per sample at the top of Section 3.2. We also provide information on the composition of the core community in the text, in section 4.5, and have clarified this text.

Line 344: the 10 most abundant taxa belonged to 8 different families? This seems improbable. How are you defining taxa here?
Yes, this is correct: the top 10 ASVs belong to these families. We have switched "taxa" for "ASVs" in the text, to ensure clarity. We chose the family level here because we could identify all ASVs to family level, but only some to genus level.

Line 348: I would still like to see the number of ASVs
Please see response above. We now provide rarefaction curves in the appendix.

Line 350: This is really only reflecting differences between the two rivers, since Bow was excluded. However, if rivers are distinct from each other, and differ by year, should samples be merged? My feeling is that this would be a major reason that the separate effects of individual streams and years needs to be accounted for.
As described above, we have moved forward with a mixed effects model for the chemistry data. At this point in the text, we are referring to the outputs of an NMDS analysis, which is well suited to exploring these types of spatial and temporal differences within the context of microbial community analyses, particularly with respect towards our research objectives within the framework of proximity to respective glaciers and season.

Line 354: To see if environmental variables explain variability is poor justification for conducting an analysis. Please expand on this. Also, how was the % variance adjusted?
The use of RDA to explore how species composition is structured across environmental gradients is quite common in ecology. To ensure our meaning is clear, we have re-worded slightly as (or, would welcome further clarification, too):

*We used RDA to assess how ASV composition varied across environmental gradients, to better understand the role of environment in structuring ASV assemblages at each of our sites.*

The suggested addition of hypotheses at the end of the Introduction should help with this too, by making the reasoning a bit more self-evident.

Percent variance was adjusted using the "RsquareAdj" function in the *vegan* package of R. This function uses a permutation method (default 1000 permutations) from Peres-Neto et al. (2006). We have added this detail to the text.

line 375: I would rather see hypotheses led comparisons rather than throwing everything at it and seeing what sticks...it really makes a difference for the reader in terms of focusing on particular results. Also, Im not sure that these are 100% independent datapoints (they have spatial and temporal autocorrelation), yet there comparisons are assuming that they are. I think this really needs to be justified that it is the best approach to show what you want to show. My feeling is that many of the points are already obvious (e.g. headwater indicators being related to temperature. Etc)
Following comments from both reviewers, we have removed this analysis. We agree that it does not add much to the manuscript.

line 377: can you say Microbacteriaceae sp? (like family sp.?) To be honest Ive never seen it before, but might be more clear to say its a species from the family Microbacteriaceae
With changes in response to the comments above, this text has been removed.

line 382: speaking of autotrophs, were you able to quantify stream turbidity or otherwise estimate algal biomass? Could help with some of the interpretations….
Unfortunately we don't have these data.

Line 426: why is this paradoxical?
In most environments, microbial-origin OM would be expected to be ~modern, so this is what we were trying to get at here. However, with edits to the manuscript, this text has been removed.

Line 430: I interpreted this slightly differently…..early in the season, the subglacial flowpaths are inefficient and poorly formed, whereas in the peak melt there is an efficient subglacial channel where most of the supraglacial melt is routed….. I think this needs to just be re-worded to make it more precise…..also, the lack of variability in this study might be due to an inadequate sample size from baseflow conditions?
We now specify that "different OM sources" refers to subglacial sources at this point in the manuscript. It is true that we had many more "melt season" samples than from outside this season, but the number of pre- and post-melt sample points (melt n = 177, pre- and post-melt n = 38) should still be enough to pick up differences in composition.

line 436: Yes, I am wondering how relevant some of these papers on Greenland are for these smaller glaciers…..not that they are bad to cite, but maybe just good to include some smaller ones as well
We have added a reference to Spencer et al. (2014), who similarly found that protein-like fluorescence increased from pre-melt to melt, for the Mendenhall Glacier (Alaska).

line 468: although there are a lot of other photoautotrophs in the streams besides cyanobacteria, such as Hydrurus, that wouldnt show up in the 16S
Yes, we certainly agree. Unfortunately we don't have 18S data, so use the presence of cyanobacteria, at least, to reinforce the point we're making.

line 490: I think priming in streams is a bit of a debate in general
Agreed - we hope that our similar understanding is reflected in our text.

Line 504: Indeed many of these are found in glacier ecosystems elsewhere, but again phyla are very broad categories. Better would be to identify groups at lower taxonomic levels where more relevant comparison can be made.
At this point in the paper, we chose this taxonomic level to enable direct comparison with the papers that were cited. We have added additional details at the order level, following similar additions to the Results.

512: Do you think that increases in alpha diversity are a response to changing environmental conditions or reflecting a greater number of cell sources? Since the mixing and residence time of streams is relatively short, my guess would be that these are not actually communities but assemblages. It is unlikely from my opinion that they are responding to their environment per se, but more that the assemblage is being formed by the inputs from the surround adjacent landscape. I think the Wilhelm paper was primarily about benthic communities.
Yes - agreed, and we now articulate this better in the paper; while our initial hypothesis was that microbial assemblages would respond to environmental gradients, it is clear from our results that this is not occurring (or, at least not to a great extent), and that downstream, an increased variety of hydrologic inputs enables greater assemblage diversity.

Considering the future, one conclusion, then, would be that the relative importance of these different inputs will change, as will their geographic location (e.g., upward migration of forests).
As articulated above, we have switched our terminology from "community" to "assemblage" as appropriate.

Line 522: what are the various lines of evidence? I think you should specifically give the argument….if these are the results from the RDA, keep in mind that this is correlative and not necessarily indicating causation….in this case that organisms prefer or are repelled by a given parameter
Here, we were specifically referring to the RDA and indicator species analyses that are discussed in the few sentences that follow. Overall, this text has changed substantially following the reworking of this section, and the referenced text has been deleted.

Line 552: I know that it can include any rank, but when I think of the word 'taxa', I am generally thinking of something at the species or ASV level. To me its really weird to refer to these much higher order taxonomies as 'taxa' when you could instead say 'phyla' 'orders' or 'classes'. I also wonder how much biological sense it makes…..For some of these its almost like referring to 'insects' or 'mammals'.
Throughout the manuscript, we have switched our terminology from "taxa" to ensure specificity. At this point in the text, we have clarified that we are describing family level or lower.

Line 565: I think a big difference is that lakes and soils are mostly stationary, while streams by definition are in motion….much more likely to get mass effects in the water column that way, and very difficult for communities to develop
Yes, of course - this is the point we are trying to convey at this point in the manuscript, and have edited to clarify our meaning.

Figure 1: Just like for the binning of longitudinal distances, the binning of the melt period also seems a bit arbitrary. For example, there is only one sampling event that seems clearly to be during a baseflow period. Also, the post melt sampling point in 2020 is associated with the third highest discharge in that given

month…..why would this not be associated with the 'melt season'? Furthermore, the sampling of pre/post season samples is really limited in comparison with the melt season samples. Also, how different is post melt to premelt…...could these just be lumped together? What is the rationale to keep them separate (like do you expect distinct patterns in the post melt vs pre melt?)? On the other hand, some of the pre-post melt samples seem that they could very easily belong to the "melt" season, yet are separated by seemingly arbitrary dotted lines in the hydrograph, as the dotted lines appear to be in a different position in 2021 than in 2019 and 2020. Thus, it seems like almost all the samples are taken during the 'melt' period based on the hydrographs. Might it make more sense to derive continuous hydrological variables rather than try to make three categories? Potentially here also continuous variables may also help with interpreting differences in the data, as I can imagine that peaks and troughs within the peak melt season are likely to also have important differences just like at different times of the year? I have to admit, I don't have a better idea, but Im also not convinced that the current strategy is the best one

Thanks for this comment. A few specific responses and points of note:

- There were some sampling points that were are not on the original Figure 1, because they were outside of the hydrograph as measured by the Water Survey of Canada. This was an error in data visualization on our part; we've modified the figure to show these (December 2019, and January 2021).
- This comment about the post-melt 2020 sample is reasonable. This sampling period followed a late warm period, after which temperatures fell rapidly. It has been switched.
- It's true that we could use discharge at this station (which is ~ 2 km downstream of the terminus of Sunwapta glacier) as an input to our mixed effects model, but that misses that - as the reviewer points out above - we are sampling a series of rivers from three (albeit proximate) glaciers (two icefields). Thus, we prefer the slightly less specific binning approach to acknowledge that there may be some difference in peaks / troughs in flow between glacier outflows, but that the overall hydrologic season "bins" should be reasonably consistent at these proximate locations. Certainly, we didn't want to neglect flow altogether in our analyses.
- The dotted lines weren't arbitrary; they were determined by the 1 $m^3$ cutoff, and so varied in timing by year. We have modified the plotting of the figure to switch to using different symbols for the three melt periods.

Figure 2: In the box and whisker plot, there is a big white gap where the samples are missing from the COVID period. While I think it is good to be transparent about missing data, it also seems weird to have a big empty spot in the middle of the figure. Would it be possible to just cut this portion out? Also, what happened to the post-melt samples…..were they combined with premelt samples?

Figure 2 has change with the move to a mixed effects model, and thus changes in plotting.

Figure 3: Was deuterium excess different among the three streams as a function of distance? I am just wondering how reasonable it is to lump the sites together in the analyses and figures. Also, while I appreciated that the season was considered in the graphing, it is really hard to pick out any seasonal patterns in these figures based on season, given that the shapes all kinda blur together at some point, and most of the trend seems to be longitudinal.

See figure S1 for trends in d-excess with distance downstream. With this figure, we are illustrating how DOC and DOM composition vary with putative water source. The new mixed effects model for d-excess shows a significant effect of distance (as a continuous variable) and season, with year and river controlled for as a random effect.

**Reviewer 2:**

This study tests how the loss of glaciers will change the composition of organic matter in downstream waters and subsequently microbial community structure. The research question is important given the rapid rate at which glaciers are being lost globally, so should be of widespread interest. The authors, somewhat unsurprisingly, discover clear shifts in the composition of organic matter along the river networks, though the magnitude of the effects on microbial community structure are small. The latter though is somewhat concerning as a negative result and begs the question whether the "right" independent and dependent variables were measured. Nonetheless, I think the paper is well put together.

The technical approaches are sound and well explained, especially the field sampling. However, the data are not statistically independent. There is spatial and temporal autocorrelation in the sampling design, and I do not think that the three-way ANOVAs consider these effects. For example, when estimating responses across distance bins, bins can show more similar values just because they are closer together in space and that needs to be accounted for in the statistical models. Although one could argue that including distance as a fixed factor would enable two closely related bins to have similar values the important point is that we can't disentangle if that effect is purely because of distance or autocorrelation. In other words, assume headwaters and near sites have similar values – is that because each site downstream is similar to its nearest upstream site (spatial autocorrelation) or because there is something special about those distance bins? The same arguments could be made for year and hydrological periods. While the breadth of field sampling is impressive, it is quite complex statistically to analyse something of this nature correctly. Thank you for these comments. As described above, we have switched from an ANOVA approach to a mixed effects model.

Important clarifications are also required for the microbial analysis. First, were there negative and positive controls in the sequencing? These controls are particularly important given the finding that the same taxa seem to dominate the composition. There should be more evidence given to rule out that lab/field contamination could be a source for the homogenization. Second, what was the read depth and how much did it vary across samples, i.e. are normalization/rarefaction techniques required? Please add this information.
**Controls:** A field blank served as a negative control through extraction, amplification, and sequencing. We used this blank to remove ASVs that were likely due to contamination from our samples using the prevalence method with a threshold of 0.5. We did include a negative control in our library preparation steps (just MM, with sterile water subbing in for a sample) to assess contaminating events during the library preparation steps. If we did not see a band following gel electrophoresis, then we proceeded with pooling and subsequent sequencing steps. Sequencing this negative has not historically provided us with enough information to further curate our dataset, and so we did not sequence the negative control. We did not include a positive control because a positive control, for these purposes, aids in determining read coverage for major taxa identified. We have used *E. coli* to spike into some samples as a positive control in the past, and have recovered expected quantities from sequenced data. As such, we did not include it here, especially because we have spiked in PhiX quite heavily during the sequencing run (~50%), to increase the diversity of the oligotrophic pool, so the reads we do obtain will correspond with mostly the most abundant microbial taxa in the sample, as a function of this and the read coverage of the MiSeq. We can add this caveat to the work.
**Read depth:** As described, above, there was a large range in read depth, ranging to ~300,000 reads. Samples below 5,000 were removed because these samples had not reached a plateau (see Figure 2, above), indicating they are not a representative sample.
**Normalization:** We Hellinger transformed our data as it gave us the most normally distributed data. We also ran the analysis on each of rarefied, Hellinger transformed and un-transformed data and did not see large differences in the results.

In all cases, we have added this information to the methods of the paper.

[Figure]

***Figure 3:*** *ASV count frequency for rarefied (L panel) and Hellinger-transformed (R panel) data.*

**Specific comments**:

Line 29:  I don't follow how this is necessarily "complex" given the previous conclusions of a core set of species that overwhelm (mass effect) environmental gradients, suggesting it is a simple predictable outcome.
As pointed out by reviewer 1, "complex ecosystem responses" is also a little bit vague. We have reworded this sentence to describe changes in organic matter and microbial source pools, and how this may affect DOM composition and microbial assemblages across our study sites.

Line 66:  shift"s"
Edited as suggested.

Line 90:  The term "mass effects" is jargon and should be defined on first use, especially for biogeochemists that may be less familiar with this "ecological" term.
Defined.

Line 153:  What volume of water was sampled for the microbes?
We collected 2L of water for each sample, and processed through the sterivex filter until the filter clogged. Details have been added to the manuscript.

Line 155:  Not sure I follow the logic for why the microbes would change in the Bow River samples but not the carbon...  They're linked…that's the argument of this entire paper.
Carbon was filtered streamside, and so we didn't have the same concerns for these samples. We will change L142 to read "always filtered streamside" to ensure clarity on this point.

Line 265:  Please can you explain why this is necessary for this comparison.
This isn't strictly necessary, but was done for visualization purposes. We have clarified our rationale in the manuscript (and see also our response to reviewer 1, above).

Line 267:  Beta-diversity is calculated from the Bray Curtis index not the NMDS.  NMDS is simply a visualization technique.
Agreed - we have modified terminology on this point.

Line 269:  How was the perMANOVA performed?  And how were the clusters identified?
The perMANOVA was performed in the R environment, using the "adonis" function from package *Vegan*, set at 999 permutations, to assess the significance between groups (based on distance from glacier). This test was conducted on a Bray Curtis dissimilarity matrix, constructed from our original microbial community dataset.  Clusters followed the "headwater", "near" and "far" bins, which were constructed *a priori.* perMANOVA was conducted on these bins. We have added some details on the analysis at this point in the text, and have also fleshed out our *a priori* choice of bins, as described in the response to reviewer 1.

Line 270:  Why have you performed the RDA?  Please explain the biogeochemical question you are trying to test.
See also our response to Reviewer 1, above. We used RDA to assess the degree to which microbial assemblages varied across the environmental gradients that we measured, to help to explore whether assemblages were shaped by environmental drivers. We have added a short sentence to clarify in Section 2.4.3.

Line 287 and throughout:  The test statistics associated with the p-values must be reported to be reproducible.  I presume here you should have some F statistic from the ANOVAs with some degrees of freedom?
We had kept the F statistics and degrees of freedom in the supplemental tables, to avoid cluttering the main text. However, with the move to a mixed effects model this text will change; statistical results are now presented in the text, and in Tables S4 and S5.

Line 335:  I don't follow what is meant by "passive overlay".
This has been clarified in the text, for others who may be unfamiliar. We are referring to a vector that is overlain on the PCA after the ordination has been conducted, such that it is shown in ordination space, but does not affect the structure of the ordination. In this case, the intent of the PCA was to explore variation in DOM composition across sites (and secondarily, "seasons"). d-excess was then added after the fact, to explore its association with the data in PCA space.

Line 341:  Aren't these 10 phyla dominant in most rivers?  It would be useful to contextualise these results, such as through comparison with the Earth Microbiome Project.
Yes, it is true that these are typically the most dominant phyla found in most rivers. Following this comments, and comments by reviewer 1, we have added information on dominant orders to further flesh out this component of the results (and later, Discussion).

A detailed comparison to the EMP project is outside the scope of this manuscript, though perhaps something that could be done with this data in the future. While we certainly agree with the reviewer's comments about the usefulness of lower taxonomic resolution comparisons and contextualization, we note that the reason that much of our text is at the phyla level is because of our wish to make comparisons with previously published literature, which typically also presents results at this taxonomic resolution.

Line 356:  Please cite evidence showing that these parameters were highly inter-correlated.
Detailed have been added to the methods – we used a common variance inflation threshold (>10) to make this determination.

Line 374:  There are no correlation statistics given anywhere to support this claim, i.e. of a trend in the clouds shown in Fig. 10.

Figure 10 plots the Spearman correlation coefficient for the relationship being discussed on the y-axis. We have added text to Section 3.2.1 and some additional description to the figure caption to better clarify the plot.

Line 419:  But is there enough of this material in a mass-balance sense to matter?

Yes, this is a fair point, and have added a few words to address this caveat. Note that we get into this more specifically for the DOM pool at the end of section 4.4.

Line 474:  A mixing model would really be the way to get at this question and the Discussion could at the very least point to its utility.

Yes, agreed. We shied away from a mixing model, given the mathematical requirement that the number of "endmembers" be no greater than (n+1) the number of tracers; at least to enable a robust model. However, we do now point to the utility of this type of approach in the Discussion.

Line 498:  I don't think this paper tests this relationship as it cannot disentangle create from consumption of OM.  The rest of this paragraph also says little about this question and just reviews the composition of bacterial families.

Yes, we agree that disentangling this relationship is beyond the scope of the paper, and was not - in fact - one of our objectives. This issue here may actually be in the sentence that follows (*"To further explore this relationship ..."*).  We have softened the language to ensure we're not being unclear on our intended scope.

Line 524:  I think this statement overstretches.  These were statistically significant but explained very little variation.  A total of like 9% all together, so how important was each variable?  I think the discussion that follows on lines 538 is much fairer.

Thanks for this comment.  Our intent here was to: (1) not discard the importance of environment altogether (see for example the reviewer's comment at L544), but also (2) provide some bridge text that moves to L538, where we discuss the fact that mass effects are much more important for regulating the composition of these assemblages. In general, this section has changed fairly substantially in response to the comments from reviewer 1.

Line 525:  Again, what is the biological significance and effect size?

The language around this point has been softened, with overall edits to this section.

Line 544:  Or we're not measuring the "right" drivers.

Yes, indeed. Though, as review 1 points out, our suite of environmental drivers was fairly extensive.

Line 546:  Again, lack of strong "control" given the variables that were measured.  That's the problem with a negative result – is it the truth or the study?

True, there could be some environmental control that we are not assessing. We can rephrase here, by rewording as "this lack of strong environmental control *from among our suite of environmental variables*".

Line 591:  Please cite some evidence to support this statement.  It does depend on the functional redundancy within these communities.

We have softened this statement, given that - as the reviewer notes - it does depend on functional redundancy.

Figure 4: I don't follow which end member the grey box corresponds with. It looks like only part of the grey box corresponds with different end members.
The grey box corresponds with end-member 4. We have articulated how end-members are visualized (arrows, grey box, pink box) in the caption, to ensure clarity.

Figure 6: I think you should add to the caption that the percentages along the axes are explained variance.
This has been added to the caption.

Figure 8: I don't follow why there are two circles if there are three groups for distance and season – which of these variables correspond to the circles and why is being omitted? I think you are looking at distance and grouping near and headwater together but the figure should be self-contained with its caption.
This is correct - in this figure the near and headwater sites are grouped because perMANOVA found that they were not distinct, but that both differed from the far sites. Plots showing other comparisons (between rivers and years) and a table with the perMANOVA results are in the supplement. We have added this detail to the caption to ensure that the figure is self-contained.

Figure 9: I find it confusing to have three values for the variance explained, none of which match each other. So the first two axes of the RDA explain 14%, all axes together explain 25%, and all axes when adjusted for the number of predictors explain 9%. Is that correct? Is it possible to focus on one number and just be more forthright that none of the environmental predictors do a particularly good job here? As for the crosses in the centre, how come the near melt sites in the bottom-left are so far away from the taxa? Are there no unique taxa associated with them? It would be informative to see the indicator species labelled on here.
Expressing the variance explained is something that the co-authors on the paper went back and forth on a fair bit. We wanted to ensure that we were representing our outputs correctly (i.e., not overstating our results by not presenting the corrected $R^2$), but also wanted to ensure clarity. We landed on the approach in this figure, which seems like it is a bit unclear to an outside reader. We will move the $R^2$ values in the bottom right of the figure (adjusted and all axes) to the caption, where we can provide a bit more context on meaning.

The species scores' clustering near the center of the dbRDA reflects the overall poor explanatory power of the ordination. We did try different plottings (including adding the indicators to this plot), and didn't find it added to our interpretation.

Figure 11: There are ca. 85 correlations here. Are you not worried about false positives, especially given that some of these Rho values look small? Also, I don't follow how the Microbactericae - temperature correlation can be more statistically significant than the one between Beggiatoceae and temperature but have a smaller absolute rho value. I haven't checked all the other columns for similar problems.
Following this comment and the comment from reviewer 1, we have decided to remove this figure. However, we note that the correlations were directly output from the statistical package used to undertake the analyses, and so this 'offset' seems due to differences in sample size.

**Literature cited:**

Cui, X., Bianchi, T. S., Jaeger, J. M., and Smith, R. W. (2016). Biospheric and petrogenic organic carbon flux along southeast Alaska. *Earth and Planetary Science Letters.* 452, 238-246.

Peres-Neto, P.R., Legendre, P., Dray, S. and Borcard, D. (2006). Variation partitioning o species data matrices: Estimation and comparison of fractions. *Ecology*. 87, 2614-2625.

Serbu, J. A., St. Louis, V. L., Emmerton, C. A., Tank, S. E., Criscitiello, A. S., Silins, U., et al. (2024). A comprehensive biogeochemical assessment of climate-threatened glacial river headwaters on the eastern slopes of the Canadian Rocky Mountains. *Journal of Geophysical Research: Biogeosciences.* 129, e2023JG007745.

Spencer, R.G.M., Vermilyea, A., Fellman, J., Raymond, P., Stubbins, A., Scott, D., and Hood, E. 2014. Seasonal variability of organic matter composition in an Alaskan glacier outflow: insights into glacier carbon sources. *Environmental Research Letters,* 9**,** 055005

---

## Author Response (AR2)

*We would like to thank the editor and anonymous reviewer for their comments and guidance, which have substantially improved our manuscript. Responses to comments are provided below, in blue, italicized, text.*

line 97: space after "the"
*Done.*

line 105: "a priori" repeated in both sentences – suggest rewording. It should also be in italics here and throughout, I think (along with 'in situ')
*Thanks for pointing out this redundancy in wording. Edits made at this point in the text, and both a priori and in situ have been italicized throughout.*

Figure 1: text on both panel of figure 1, but especially panel A, should be enlarged, as it is almost impossible to read them. Also, perhaps it could be possible to overlay the path of a given stream with a line (maybe in different colors by stream?) so that we can see their placement on the map?
*We have re-made Figure 1 as a stacked figure, to allow for larger text sizes. Hopefully this enables the lines denoting the streams to be a bit clearer, too.*

line 236: maybe explicitly state here what samples are being pooled? It's a bit confusing as written
*This is now clarified in the text: Any samples showing an expected band size of ~400bp were purified using a bead cleanup protocol. The cleaned bands were then pooled to make the final library for sequencing using 5 uL of each sample; if a sample was more faint than other bands, 10uL of that sample was added.*

line 244: BSA in parentheses?
*Done.*

line 243-245: seems like this info should come before the sequencing information?
*Agreed. We have moved this information to the beginning of the section to help with flow.*

line 254-257: Im not familiar with PhiX…..would it be possible to briefly explain what this is and what it does? Also, would this need to be also outlined above somewhere?
*The following information on PhiX has been added at this point in the text:*
*[PhiX] is a quality control reagent used commonly in sequencing runs to optimize cluster generation, sequencing, alignment and calibration control throughout the run. Because PhiX is a well-defined bacteriophage genome, it has a diverse base composition that provides the balanced fluorescent signal that low diversity sample libraries, like ours, lack, during each sequencing cycle (Illumina).*
*We also add a reference to this added text where PhiX is briefly mentioned, above.*

line 257: Although 5,000 is not terribly many at the end of the day, so would probably make sense that they do not reach a plateau at this point? What kinds of range did you have with your sequence numbers per sample? Lastly, be careful in calling these ASVs 'species'
*Yes, agreed. We have added a reference to the supplemental figure that shows rarefaction curves to give the reader a sense of range of sequence numbers per sample. Thanks for pointing out that we use species erroneously here: this has been changed to ASVs.*

line 289: were alpha diversity measures calculated on rarefied data? And what was the rarefaction level? I saw that you threw out everything below 5000, so was it 5000? this should be explicitly stated somewhere (although I realize that rarefaction may not be necessary if all samples plateau, in which case im not sure why rarefaction is mentioned here? just some clarification necessary)

*Alpha diversity was calculated on untransformed data; this has been clarified in the text. We double checked our text on this point; rarefaction only comes up at the end of this paragraph, where we state that we compared our Hellinger-transformed NMDS and perMANOVA analyses using a rarefied dataset. We've added a few words for clarity at this point in the text.*

line 294: I thought that the mid sites were excluded from the analysis?

*Thank you, this is a typo. Mid should read near and has been changed.*

line 323: headwater and near sites? Try to keep all terminology similar throughout the paper so that the reader can keep it all straight

*This is correct as written: our distance bins are headwater, near, mid (for chemical analyses only), and far. No edit made at this point in the text.*

line 365: A-ha….here are the read details….might they be better above? I leave this to the discretion of the editor and authors. Also, is 'Rarefaction curves' a proper noun?

*Agreed that these details are better in the methods section. We have moved and integrated this section into lines 321-323 (tracked changes version). We have also removed the capitalization for "rarefaction".*

line 378: might it be worthwhile to give the genera names for some of these common ASVs? Family names (for example) do not provide a lot of information, although its better than nothing of course

*Thank you for your comment. For amplicon sequencing runs such as ours, we are rarely able to resolve down to the genus level, even when using the fairly high confidence threshold of 0.8, when aligning our sequences to the taxonomy database. This kind of reporting is usually rare in 16S environmental publications, especially for oligotrophic environments such as ours. We also lose some confidence the further down the taxonomic tree we go. As it is, resolving down to the family level is usually the best we can do. Indeed, only 2 of the top 10 families actually resolve into genera. For this reason, we will not specify genera further for these top 10 families.*

line 382: would it be possible to give some hard numbers on the diversity values here? Its great to know that they change from upstream to downstream, but neither the text nor the figures really gives the reader a sense of the magnitude of change, or what is there in the first place.

*Thank you for the suggestion. We have added median diversity values pre and post-melt to illustrate the difference in alpha diversity between near and far sites throughout the melt season.*

line 402: have you used this ISA acronym before?

*Yes, ISA (indicator species analysis) is first defined at L304. No change made at this point in the text.*

line 405: again, be careful in the use of 'species'

*Thank you; changed in both cases to ASV.*

line 451: consistent use of 'carbon' versus "C"

*C is switched to "carbon"*

Figure 7: In panel A, both the Far sites and the intersection of Far sites with Headwaters have the same values (73.94%, 306,996).....this is a mistake, no?
*Yes, this is a typo – thanks for catching our error!  It has now been corrected (now Figure 6).*

line 540: Again, while true, these are pretty coarse/vague taxonomic entities
*Yes, we agree, but we purposefully retained this text in the last iteration to allow for broad comparison with other studies, which often present results at this taxonomic resolution. Later in the paragraph, we provide results at a  finer taxonomic resolution. No edit made.*

line 597: should mass effects be explicitly defined?
*We have added a definition as "homogenization by high rates of dispersal", similar to the definition provided in the abstract.*

line 614: assemblages rather than communities?
*We have replaced community with assemblage at this point in the text, and have done one last "find and replace" to ensure consistency throughout.*

Conclusions: this sections seems a bit long for me for a conclusions section – also don't know if references are appropriate here, but will leave that to the discretion of the journal
*We have left the conclusions section as-is, but are happy to take further suggestions on length or suitability of references.*

Figure 2: match letter cases with text and figure (i.e. 'a' versus 'A')
*This has been changed in the figure caption.*

GENERAL COMMENT ON FIGURES: if there would be any way to create some sort of color / shape scheme that would be possible to unify across figures, that would greatly aid in interpretation for the reader. Right now, each figure really has its own key in terms of what color / shape means, and it is hurting my head a bit going from one figure to the next
*Yes, one unfortunate side effect of the move to the mixed effects models was that we had to expand on our previously unified colour scheme.  However, we have put some thought into this, to make sure that our scheme was consistent, with different base colours to represent the different factors that we sample across, as appropriate for the figure in question.  Our scheme is as follows:*

- *Our figures primarily differentiate near/mid/far sites, which was our original schema. For these plots (Figures 1a, 4-9), we use colours across a range of green, tan, and brown, often with symbol shape differentiating between hydrologic periods. Notably, we retained this plotting for all microbial plots, and for plots where binning was useful to show distance downstream, but analysis was not via a mixed effects model (compositional PCA, 13C-14C biplots)*
- *With the move to the mixed effects model and edits implemented following the initial review, we moved some plots away from distance bins, to enable plotting of data along a distance gradient. This approach was implemented for all plots associated with a mixed effects model. Here, the plotting scheme de-emphasizes river and year (random effects in our model).*
    - *For Figure 2, we plot to emphasize fixed effects either on the x-axis (distance) or via faceting (hydrologic period across columns), and then use colours (river) and symbols (year) for the random effects.*
    - *For Figure 3 (which now combines previous figures 3 and 5) we plot to emphasize distance (x-axis), and use colour (rather than faceting) to illustrate hydrologic period. Figure 1b uses this same colour scheme to show hydrologic period.*

Figure 3: would it be possible to break this figure up by river (i.e. make separate panels by river)? Right now there is so much going on that its really hard to see any pattern. If there is not a pattern you want to highlight, could alternatively merge with figure 5 to create a new panel?

*We've chosen to not break up by river, because we control for this (as a random effect) in our models. Given that we don't focus too much on the original figure 3 in the text, we now combine both Figure 3 and Figure 5 into a new, 3-panel Figure 3.*

Figure 4: these are for all sites and seasons combined? Should specify this…

*Yes. This is now specified in the figure caption.*

Figure 8: what exactly are the different groupings shown by the circles? Is there any way to show river name as well? 'holms' should probably be capitalized, and the 2 in R2 superscript

*(Now Figure 7). We've tried to clarify further in the caption: circles show that the far sites differ significantly from the headwater and near sites, but that headwater and near do not differ from one another.*

*Plotting by river (and, also year) is shown in the Supplement, and we now point the reader to these plots in the figure caption.*